# ONLINE VERSATILE INCREMENTAL LEARNING: TOWARDS CLASS AND DOMAIN-AGNOSTIC ADAPTATION AT ANY TIME

## ABSTRACT

Continual learning enables vision systems to adapt to ever-changing data distributions. Despite significant advances, existing approaches fail to capture the seamless, concurrent transitions, a critical capability for real-world deployment. This work introduces **Online VIL (Online Versatile Incremental Learning)**, a novel scenario where class concepts and visual domains evolve simultaneously online without explicit boundaries. To better adapt to the challenges of such dynamic environments that more closely resemble real-world conditions, we propose a novel framework **TopFlow**, **Top**ology preservation with **Flow** matching representation that contains two complementary mechanisms: **Domain-agnostic Flow Matching (DFM)** and **Global Topology Preservation (GTP)**. DFM guides the model to have domain-agnostic representations by integrating the geodesic flow kernel into contrastive learning. In contrast, GTP maintains the global structure of the feature space without explicitly storing past examples. Our extensive experiments demonstrate that TopFlow effectively addresses the limitations of existing methods within the Online VIL scenario, achieving state-of-the-art performance in challenging Online VIL. The proposed methods suggest potential directions for building continual learning systems in realistic dynamic environment.

## 1 INTRODUCTION

Continual Learning (CL) (Li & Hoiem, 2017a; Kirkpatrick et al., 2017; Wang et al., 2022a;c;b; Smith et al., 2023; Zhang et al., 2023; Park et al., 2024a) has gained increasing attention as deep learning moves closer to the real world, where data distributions evolve. A key challenge is catastrophic forgetting, where adapting to new knowledge disrupts previous knowledge. To mitigate this, prior research has introduced distinct paradigms such as Class Incremental Learning (CIL) and Domain Incremental Learning (DIL), each targeting a specific form of non-stationarity. More recently, Online Continual Learning (OCL) (Wang et al., 2023a; Wei et al., 2023; He et al., 2024) has been proposed to handle streaming data under memory and single-pass constraints, including scenarios with blurry task boundaries (Koh et al., 2021; Bang et al., 2021; Moon et al., 2023). However, most studies still rely on CIL or DIL, limiting their ability to capture the full complexity of real-world dynamics.

A recently introduced Versatile Incremental Learning (VIL) (Park et al., 2024b) suggests a more realistic scenario where new tasks have a broader chance to evolve in both directions of the classes and domains, without prior knowledge. While VIL marks a divergence from realism, it assumes discrete increments provide implicit structural information about distribution shifts. In contrast, real-world environments exhibit continuous transitions without boundaries, offering no such organizational cues. Moreover, environmental change in reality is continuous and unpredictable, due to locally constrained information and multi-factor interactions. For instance, an autonomous driving system may suddenly face both a new object and a shift in weather or city conditions, requiring immediate (online) adaptation without knowing whether it concerns classes, domains, or both.

To this end, we introduce **Online Versatile Incremental Learning (Online VIL)**, a new scenario that captures both the unpredictable heterogeneous shifts in an online manner. Online VIL is distinguished by reflecting the evolution of the natural information stream, characterized by unpredictable, gradual, and heterogeneous shifts that occur along multiple evolutionary trajectories. As shown in Figure 1,

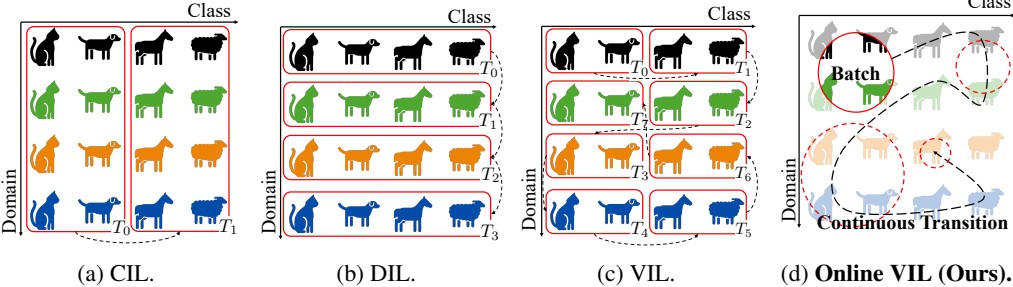

Figure 1: Conceptual comparison of (a) Class Incremental Learning (CIL), (b) Domain Incremental Learning (DIL), (c) Versatile Incremental Learning (VIL), and (d) Online Versatile Incremental Learning (Online VIL, Ours). Red lines indicate the scopes observable at once, while black dashed lines depict transitions across time.

Online VIL ensures a degree of freedom for transition; each sequence presents distinct challenges without heuristic patterns. Consequently, Online VIL enables faithful evaluation of CL models and provides a foundation for systems that operate under real-world dynamics.

In the Online VIL scenario, adaptation to chaotic shifts in classes and domains without explicit access to prior inputs is crucial for distinguishing class-discriminative knowledge from domain-specific knowledge. Otherwise, models rely on the spurious features of current distributions and tend to exhibit rapid forgetting and a lack of generalization. Through systematic layer-wise feature analysis of pre-trained Vision Transformers, we observe that early layers predominantly capture domain-specific patterns (e.g., texture, lighting), while deeper layers encode class-discriminative features and more abstract semantic representations. This motivates a novel **Domain-agnostic Flow Matching (DFM)** technique, which aligns features with the intrinsic geometry of pre-trained knowledge through reconceptualized geodesic flow kernel (Gong et al., 2012) while mitigating domain-specific shifts.

In addition to class and domain-agnostic alignment, preserving the global topology of learned features is essential for maintaining semantic continuity across evolving tasks. Conventional objectives often fail to maintain the structural relationships of feature space because they shrink the occupation of missing classes in the feature space. To address this, we propose **Global Topology Preservation (GTP)**, which preserves invariant geometric configurations in feature space, enabling robust knowledge retention without requiring complete class coverage.

Integrating these components, we introduce **Top**ology preservation with **Flow** matching representation (**TopFlow**), a novel framework designed for Online VIL. With recognition of DFM and GTP regularization for geometry, TopFlow ensures robust adaptation to unpredictable shifts. To evaluate its effectiveness, we conduct extensive experiments in Online VIL and observe that TopFlow consistently outperforms existing state-of-the-art methods across multiple benchmarks.

Our contributions are summarized as follows:

- We introduce **Online VIL**, a realistic scenario for evaluating continual learning under continuously evolving both class and domain distributions with ambiguous task boundaries.
- We reveal a novel role of the pre-trained ViT layer that encodes class and domain knowledge. Leveraging this insight, we propose Domain-agnostic Flow Matching (**DFM**) to learn domain-agnostic representations by integrating the geodesic flow kernel into contrastive learning.
- We propose Global Topology Preservation (**GTP**), a mechanism for maintaining knowledge representations using feature topologies without explicit memory of previous inputs.
- We demonstrate that the proposed **TopFlow** significantly outperforms existing state-of-the-art methods through comprehensive experiments in our challenging Online VIL scenario.

## 2 RELATED WORK

### 2.1 ONLINE CONTINUAL LEARNING

Online Continual Learning (OCL) (De Lange et al., 2021; Gunasekara et al., 2023) has emerged as a pragmatic paradigm that reflects the challenges of real-world settings, where data arrive as continuous

streams and models must operate under minimal batch sizes, single-pass constraints, and strict computational and memory limitations. Traditionally, OCL methods rely on replay buffers (Rolnick et al., 2019; Mai et al., 2021) to store a small subset of past data, thereby mitigating catastrophic forgetting while learning new tasks. Recent advances explore leveraging prototypes (Wei et al., 2023), replay-free strategies (Zając et al., 2024), and pre-trained models with prompt (Moon et al., 2023). While effective in class or domain increments, this dependence on memory limits scalability and realism. Methods addressing more realistic scenarios with blurry or ambiguous task boundaries have emerged (Koh et al., 2021; Bang et al., 2021). However, the replay methods require growing memory in proportion to task diversity. Also, blurry boundary methods still assume either class-only or domain-only shifts, failing to capture the heterogeneous evolution of a realistic data stream.

Therefore, we propose Online Versatile Incremental Learning (Online VIL). This scenario exposes models to unpredictable shifts in both classes and domains while enforcing online constraints that limit memory and multi-pass access.

## 2.2 Geodesic Flow Kernel

The Geodesic Flow Kernel (GFK) (Gong et al., 2012; Gopalan et al., 2011) has been widely used in unsupervised domain adaptation to align feature distributions between a predefined source and target domain. Conventional applications approximate the geodesic on the Grassmannian manifold between two static domains, which requires that the data of the source and target domains are fully available and static. In contrast, Online VIL presents unique challenges: domain shifts occur continuously and unpredictably, and data arrive sequentially, making it impossible to estimate the flow offline.

To address these challenges, we reformulate the approach from GFK and design a novel Domain-Agnostic Flow Matching (DFM). Unlike traditional geometry estimation in feature space, DFM is designed for sequentially arriving data and evolving domains without relying on holistic data access. Through both empirical and theoretical analysis of feature geometry, DFM enables online alignment on feature geometry in dynamic conditions, avoiding computationally expensive higher-order manifold computations. This design fundamentally extends the applicability and purpose of GFK, enabling robust domain-agnostic feature matching in the OCL.

## 2.3 Feature Topology

Several works have leveraged the topology of feature space to mitigate catastrophic forgetting in sequential learning. Tao et al. (2020a) employs elastic Hebbian graphs to preserve neighborhood relationships during incremental updates, while Tao et al. (2020b) uses self-organizing maps to identify representative feature points and restrict their displacement. More recent approaches, such as Liu et al. (2022) and Wang et al. (2023b), maintain pair-wise instance similarity or local topological relations by decomposing the global structure, further reducing forgetting across tasks.

However, these methods are limited to offline CL, requiring full access to past samples to construct the feature topology. Furthermore, existing topology preservation methods rely on complete semantic information across all classes. In online scenarios where only partial class information is available at each time step, these methods suffer from incomplete topology construction, resulting in suboptimal feature space organization. In contrast, our work proposes a Global Topology Preservation (GTP) strategy that maintains structural knowledge without requiring the storage of previous data.

# 3 Method

## 3.1 Problem Setup: Online Versatile Incremental Learning

In real-world scenarios, data distributions gradually evolve, often exhibiting significant variability across multiple dimensions such as spatial domains, semantic classes, and temporal shifts. We propose a novel scenario termed Online Versatile Incremental Learning (Online VIL), which simulates these properties by constructing a continuous data stream with stochastically varying distributions.

Given a dataset with $n_{\mathcal{D}}$ domains and $n_{\mathcal{C}}$ classes, we define the product category space as follows:

$$\mathcal{X} = \bigcup_{i=1}^{n_{\mathcal{D}}} \mathcal{D}_i, = \bigcup_{j=1}^{n_{\mathcal{C}}} \mathcal{C}_j, = \bigcup_{i=1}^{n_{\mathcal{D}}} \bigcup_{j=1}^{n_{\mathcal{C}}} \mathcal{K}_{i,j}, \quad \text{where} \quad \mathcal{K}_{i,j} = \mathcal{D}_i \cap \mathcal{C}_j, \tag{1}$$

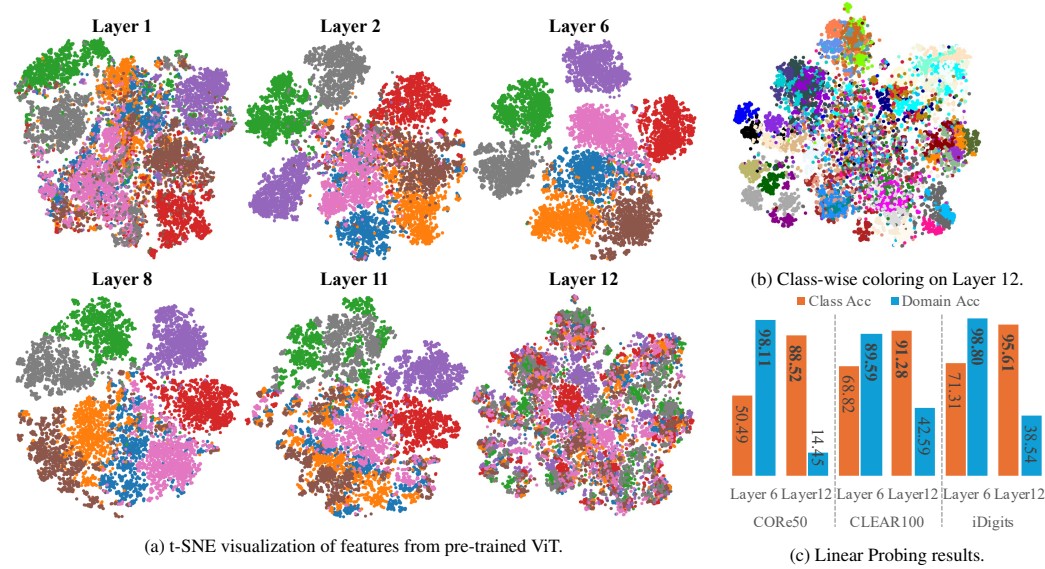

(a) t-SNE visualization of features from pre-trained ViT.

(b) Class-wise coloring on Layer 12.

(c) Linear Probing results.

Figure 2: Analysis of layer-wise knowledge on pre-trained ViT for data with changing distributions (CORe50). (a) t-SNE visualization of features from certain layers of the pre-trained ViT. The color indicates the domain. (b) The t-SNE visualization of the last layer with class-wise color. (c) Accuracy of linear probing for class and domain classification from different layers.

where $\mathcal{D}_i$ is a set of samples in a domain $i$ and $\mathcal{C}_j$ is in a class $j$, and $\mathcal{K}_{i,j}$ is samples that belong to domain $i$ and class $j$. Machine learning typically assumes the data $\mathcal{X}$ as a lower-dimensional manifold embedded in a high-dimensional space Cayton et al. (2005). In this context, we can conceptualize the data stream as a trajectory through the joint domain-class space, where each timestep $t$ provides an observation window $\mathcal{V}_t \subset \mathcal{X}$ which is a disjoint open neighborhood of data $\mathcal{X}$. This geometric interpretation naturally leads to our manifold-based approach in the subsequent technical development. Inspired by Si-Blurry (Moon et al., 2023), we divide categories into disjoint sets ($\mathcal{K}_{\text{disjoint}}$) with clear distributional boundaries and blurry sets ($\mathcal{K}_{\text{blurry}}$) with deliberately abstracted distributions.

Algorithm A.1 in Appendix details our task construction process, which proceeds in six main steps: (*i*) *category partitioning* to establish variable distributional clarity, (*ii*) *sample extraction* to create diverse distributional patterns, (*iii*) *sample redistribution* to simulate realistic category overlap, (*iv*) *randomized task assignment* to eliminate artificial task boundaries, (*v*) *task construction* with varying characteristics, and (*vi*) *batch generation* with constrained visibility windows. The Online VIL scenario distinguishes itself from traditional CL scenarios through three key characteristics:

1. **Locally Limited Visibility.** At any time step, the model observes only a small fraction in both number of samples ($|\mathcal{V}_t| \ll |\mathcal{D}|$) and categories ($\exists \mathcal{K}_{i,j} : x \in \mathcal{K}_{i,j} \land x \notin \mathcal{V}_t$), reflecting real-world constraints on data accessibility, combines both spatial and temporal restrictions.
2. **Continuous Smooth Variation.** The variation is smooth and continuous over time, making it hard to distinguish the distribution shift without a global context.
3. **Dynamic Distribution.** Each category appears and disappears gradually. The variance across spatial (domain appearances), semantic (class properties), and temporal (distribution evolution) dimensions requires simultaneous adaptation to new patterns and retention of previously acquired knowledge.

This formulation demands adaptation mechanisms that extract meaningful patterns despite high variance and constrained observability. Traditional CL methods, which assume either domain stability or class stability, fail to resemble these fundamental challenges in real-world perception systems.

### 3.2 DOMAIN-AGNOSTIC FLOW MATCHING

The Online VIL scenario, with limited visibility, multi-dimensional variance, and dynamic distribution, poses challenges beyond traditional CL. While the original VIL work measured feature similarity, it did not examine how domain and class signals are encoded in the backbone under noisy

class and domain shifts. This is critical because the one-pass constraint and lack of rehearsal in the online regime can make representations unstable.

To better understand this, we analyze the representations of frozen pre-trained Vision Transformers, which have become the standard backbone in recent CL research (Wang et al., 2022a;b;c; Smith et al., 2023; Gao et al., 2023). Although not trained under Online VIL, the frozen backbone provides a stable proxy, reflecting common practice in CL of freezing the ViT and updating only a small number of parameters. Its knowledge largely dictates how domain-specific variations and class-level semantics interact. Motivated by this, we conduct a layer-wise analysis, expecting deeper layers to align more closely with class semantics, as their outputs directly drive the classification head.

We perform a t-SNE study and linear probing to investigate this. Our t-SNE visualizations (Figures 2a, 2b) show that intermediate features group by domain, while final features group by class. Linear probing (Figure 2c) further confirms stronger domain discrimination in intermediate layers and stronger class discrimination in the final layer. These findings reveal a structural representation gap in Online VIL, motivating our proposed Domain-agnostic Flow Matching (DFM), a geometry-informed contrastive loss designed to learn domain-agnostic representations.

The Geodesic Flow Kernels (GFK) (Gong et al., 2012) suggest that an assumption can be made that features from each domain compose a subset of the Grassmann manifold. Our approach innovates this concept by acquiring the geometric relationship between layers rather than inter-domain, which may be impossible to access in the continual learning scenario. For convenience, we use the function with the same dimension for each layer, like Vision Transformers (ViTs) (Dosovitskiy et al., 2020).

With a cascade of functions with residual connections:
$$\boldsymbol{h}_0 = \boldsymbol{x}, \quad \boldsymbol{h}_n = f_n(\boldsymbol{h}_{n-1}) + \boldsymbol{h}_{n-1}, \quad n = 1, 2, \ldots, l, \tag{2}$$
where $f_i : \mathbb{R}^{b \times d} \to \mathbb{R}^{b \times d}$ is a function of the $i$-th layer, and $\boldsymbol{h}_l \in \mathbb{R}^{b \times d}$ is the last layer feature with batch size $b$ and feature dimension $d$. With local neighborhood $\mathcal{V}_t$ on a Riemannian manifold as mentioned Section 3.1, the infinitesimal derivation of feature $\boldsymbol{h}_l$ can be expressed with infinitesimal variation on the tangent space $\delta \boldsymbol{h}_{n-1}$:

The Geodesic Flow Kernels (GFK) (Gong et al., 2012) suggest that an assumption can be made that features from each domain compose a subset of the Grassmann manifold. Our approach innovates this concept by acquiring the geometric relationship between layers rather than inter-domain, which may be impossible to access in the continual learning scenario. For convenience, we use the function with the same dimension for each layer, like Vision Transformers (ViTs) (Dosovitskiy et al., 2020).

With a cascade of functions with residual connections:
$$\boldsymbol{h}_0 = \boldsymbol{x}, \quad \boldsymbol{h}_n = f_n(\boldsymbol{h}_{n-1}) + \boldsymbol{h}_{n-1}, \quad n = 1, 2, \ldots, l, \tag{3}$$
where $f_i : \mathbb{R}^{b \times d} \to \mathbb{R}^{b \times d}$ is a function of the $i$-th layer, and $\boldsymbol{h}_l \in \mathbb{R}^{b \times d}$ is the last layer feature with batch size $b$ and feature dimension $d$. With the local neighborhood $\mathcal{V}_t$ on a Riemannian manifold as mentioned in Section 3.1, we can take a first-order approximation with the infinitesimal variation of feature $\boldsymbol{h}_l$, as detailed in Equation A.3-A.5. Then the inner product between $\boldsymbol{h}_n$ and $\boldsymbol{h}_l$ as:

$$\langle \boldsymbol{h}_n, \boldsymbol{h}_l \rangle = \int_X \boldsymbol{h}_n^T \boldsymbol{h}_l d\boldsymbol{x} = \int_X (\bar{\boldsymbol{h}}_n + \delta \boldsymbol{h}_n)^T (\bar{\boldsymbol{h}}_l + \delta \boldsymbol{h}_l) d\boldsymbol{x}$$
$$= \mathbb{E}_X \left[ (\bar{\boldsymbol{h}}_n + \delta \boldsymbol{h}_n)^T (\bar{\boldsymbol{h}}_l + \delta \boldsymbol{h}_l) \right] \tag{4}$$

where the expectation of feature $\bar{\boldsymbol{h}}_n = \mathbb{E}_{\boldsymbol{x} \in \mathcal{V}_t}[\boldsymbol{h}_n]$ being cotangent $T^* F_n$, and the variation $\delta \boldsymbol{h}_n = \boldsymbol{h}_n - \bar{\boldsymbol{h}}_n$ belonging to tangent space $T F_n$.

When we utilize SVD decomposition for the combined space of $\boldsymbol{h}_n$ and $H = \begin{bmatrix} \boldsymbol{h}_n^T & \boldsymbol{h}_l^T \end{bmatrix}^T = U \Sigma V^T$. As detailed in Equation A.3, we can approximate the inner product between the function spaces by projecting onto their common subspace $U$ as an orthogonal basis. Based on our empirical observation in Figure 2:

- $\delta \boldsymbol{h}_n^T \delta \boldsymbol{h}_n$ represents the residual component ($\boldsymbol{u}_{\text{residual}}$).

- $\delta \boldsymbol{h}_n^T \delta \boldsymbol{h}_l$ and $\delta \boldsymbol{h}_l^T \delta \boldsymbol{h}_n$ represent push-forward induced by transportation ($\boldsymbol{u}_{\text{push}}$).

- $\delta \boldsymbol{h}_n^\top \left( \Phi_n^l \right)^\top \Phi_n^l \delta \boldsymbol{h}_n$ provides metric of the push-forward transportation ($\boldsymbol{u}_{\text{metric}}$), curvature information with its derivative.

This decomposition provides crucial insights, as the proper selection of layer $\boldsymbol{u}_{\text{residual}}$ naturally captures domain-specific variations, while the transformation components encode semantic abstractions. In contrast, $\boldsymbol{u}_{\text{push}}$ and $\boldsymbol{u}_{\text{metric}}$ encode semantic transformations that correlate with class boundaries. Especially $\boldsymbol{u}_{\text{metric}}$ performs as a role of optimization for metric in local neighborhood, enables to delicate geometric optimizations. With this insight, we can guide intermediate representations away from domain-specific geometry and toward class-discriminative. However, explicitly isolating the basis vectors corresponding to components is non-trivial. We introduce the Domain-agnostic Flow Matching Loss ($\mathcal{L}_{\text{DFM}}$), which utilizes contrastive learning with geometry-guided representations. Let $\boldsymbol{h}_n^*$ be a perturbed function of $\boldsymbol{h}_n$ with continuous deformation; we are locally able to apply the same $U$. For a sample $\boldsymbol{x}_{(i)}$ in batch $\boldsymbol{X}$, let the intermediate feature $\boldsymbol{h}_{n,(i)} \in \boldsymbol{H}_n$, last-layer feature $\boldsymbol{h}_{l,(i)} \in \boldsymbol{H}_l$ from the frozen backbone model, and intermediate features from the perturbed function $\boldsymbol{h}_{l,(i)}^* \in \boldsymbol{H}_n^*$. We define $\boldsymbol{H}_{(i)}^+ = \left\{ \boldsymbol{h}_{l,(i)} \mid \boldsymbol{h}_{l,(i)} \in \boldsymbol{H}_l \right\}$ as positive pairs, and $\boldsymbol{H}_{(i)}^- = \left\{ \boldsymbol{h}_{n,(i)} \mid \boldsymbol{h}_{n,(i)} \in \boldsymbol{H}_n \right\} \cup \left\{ \boldsymbol{h}_{l,(i)} \mid i \neq j \wedge \boldsymbol{h}_{l,(j)} \in \boldsymbol{H}_l \right\}$ as negative pairs. We propose a contrastive loss formulated as follows:

$$\mathcal{L}_{\text{DFM}} = -\frac{1}{M} \sum_{m=1}^{M} \log \frac{\sum_{\boldsymbol{h}_{n,(m)}^+ \in \boldsymbol{H}_{(m)}^+} \exp\left( \langle \boldsymbol{h}_{n,(m)}^*, \boldsymbol{h}_{(m)}^+ \rangle / \left( \tau \left\| \boldsymbol{h}_{(m)}^* U \right\| \left\| \boldsymbol{h}_{(m)}^+ U \right\| \right) \right)}{\sum_{\boldsymbol{h}_{n,(m)}^- \in \boldsymbol{H}_{(m)}^-} \exp\left( \langle \boldsymbol{h}_{n,(m)}^*, \boldsymbol{h}_{(m)}^- \rangle / \left( \tau \left\| \boldsymbol{h}_{(m)}^* U \right\| \left\| \boldsymbol{h}_{(m)}^- U \right\| \right) \right)}, \quad (5)$$

where $\tau$ is a temperature, while dividing by norm reduces the effect of the amplitude of the feature. The further details of the derivation are provided in Appendix A.2 and Algorithm A.2.

This formulation guides the perturbed feature $\boldsymbol{h}_n^*$ to capture class-relevant information by aligning with the output feature from frozen backbone $\boldsymbol{h}_l$ while implicitly minimizing its encoding of pure domain-specific cues in $\boldsymbol{h}_n$. The DFM loss acts as a robust anchor, leveraging the geometric structure to be invariant to domain shifts while preserving class-discriminative distances. Furthermore, batch-wise computing enables the model to locally learn nuanced, domain-invariant knowledge, while permitting intermediate layers to capture necessary domain-specific adaptations under guidance.

## 3.3 GLOBAL TOPOLOGY PRESERVATION

One significant challenge in the Online VIL scenario is that the model needs to train without samples that can represent the overall distribution. This challenge is particularly acute in the Online VIL scenario, where input batches exhibit non-stationary class-domain compositions. To address this representational instability and to ensure topological coherence of the feature space, we introduce **Global Topology Preservation (GTP)**, a novel approach designed to maintain semantic structural integrity across temporally evolving data streams. The effective information content of batch $B_t$ at layer $\boldsymbol{h}_l(\boldsymbol{x})$ can be measured by the rank of its empirical covariance matrix:

$$C_t = \mathbb{E}_{\boldsymbol{x} \in B_t} \left[ (\boldsymbol{h}_l(\boldsymbol{x}) - \mu_t)(\boldsymbol{h}_l(\boldsymbol{x}) - \mu_t)^T \right], \quad \mu_t = \mathbb{E}_{\boldsymbol{x} \in B_t} \left[ \boldsymbol{h}_l(\boldsymbol{x}) \right], \quad (6)$$

When certain classes are absent, $C_t$ captures only partial information about the global feature distribution, leading to rank deficiency. When batch $B_t$ contains samples from only $k < C$ classes, the rank of the gradient is limited to $k$; the gradient provides not only the construction of decision boundaries for the present classes but also distorts the feature space by contracting around them. This rank deficiency leads to incomplete representations in the feature space, biased gradients toward observed classes, resulting in distortions and instability as the feature space contracts around present classes, while failing to maintain representations for absent classes. This issue persists even with a prototype-based classifier (Snell et al., 2017), which assumes each class is independent, thereby removing the absent rank into a zero-eigenvalue space.

To circumvent these limitations while preserving global topological properties, GTP constructs a surrogate representation of the feature manifold through two key components: a set of $k$ global prototypes $\{\bar{p}_g^j\}_{j=1}^k$, and a set of global relationship vectors $\{\bar{r}_g^{j,l}\}_{j,l=1}^k$. For each incoming batch $B_t$, we derive batch-specific prototypes $\{p_b^i\}_{i=1}^k$ from the final layer features using FINCH clustering (Sarfraz et al., 2019). We employ an exponential moving average (EMA) update strategy to ensure smooth temporal evolution of global prototypes while mitigating batch-to-batch fluctuations. Once the correspondence is established, each global prototype $\bar{p}_g$ is updated with the batch prototype $p_b$:

$$\bar{p}_g^{\pi^*(i),\text{new}} = (1-\alpha)\bar{p}_g^{\pi^*(i),\text{old}} + \alpha p_b^i \quad \text{for } i = 1, \ldots, k, \quad (7)$$

where $\alpha$ controls EMA update rate, balancing stability and adaptability. We find optimal assignment $\pi^*$ by solving: $\pi^* = \arg\min_{\pi \in S_k} \sum_{i=1}^k \|p_b^i - \bar{p}_g^{\pi(i)}\|$ using Hungarian algorithm Kuhn (1955).

The relationships between its constituent elements fundamentally characterize the topological structure of a manifold. To capture these structural properties, we introduce a learnable mapping function $\psi : \mathbb{R}^{2d} \rightarrow \mathbb{R}^m$, where concatenation preserves both individual prototype information and their relative positioning. between batch-specific relationship vectors $r_b^{i,j} = \psi([p_b^i; p_b^j])$ and global relationship vectors $\bar{r}_g^{i',j'} = \psi([\bar{p}_g^{i'}; \bar{p}_g^{j'}])$. The GTP loss then enforces alignment between these relationship structures as follows:

$$\mathcal{L}_{\text{GTP}} = \sum_{i=1}^{k} \sum_{\substack{j=1 \\ j \neq i}}^{k} D(r_b^{i,j}, \bar{r}_g^{\pi^*(i),\pi^*(j)}), \tag{8}$$

Where $D$ represents the cosine distance metric, this formulation encourages consistent pairwise relationships between semantic prototypes, effectively preserving the global topological structure without explicitly computing spectral properties.

Since the number of global prototypes $\bar{p}_g$ is smaller than the number of samples in each batch, these prototypes deliberately capture a coarse-grained representation, reflecting the intended vagueness under limited observations. Each prototype is updated considering its relationships with all other prototypes. These prototypes cover the feature space of several semantic classes, which reduces the distortion of the feature space from the concept without samples. Additionally, the EMA update strategy stabilizes the prototypes over time and removes outdated information from previous batches. As shown in Figure A.3 and A.4, this effectively filters out the unseen class information in recent batches, performing a role similar to a short-term memory. This mechanism provides a computationally efficient solution to maintaining representational stability in non-stationary learning environments, complementing the domain-invariance properties induced by DFM.

## 4 EXPERIMENTS

This section comprehensively evaluated and compared our proposed methods against state-of-the-art approaches on established benchmark datasets. Section 4.1 describes the experimental setup, including datasets, comparison baselines, and evaluation metrics. Section 4.2 presents extensive quantitative results, demonstrating the effectiveness of our method across multiple benchmarks. To further interpret our approach, Section 4.3 provides in-depth analyses, including ablation studies, to isolate and assess the contribution of each component in our framework. Further details about the experiments are provided in the Appendix B.

### 4.1 EXPERIMENTAL SETUP

**Datasets.** We conducted experiments on three benchmarks, including iDigits (Volpi et al., 2021), CORe50 (Lomonaco & Maltoni, 2017), and CLEAR100 (Lin et al., 2021), for which it is possible to construct Online VIL scenarios that can cause a significant shift in distribution by clearly distinguishing both classes and domains. We split the CORe50 and CLEAR100 datasets into 10 tasks for the task construction, and 5 for the iDigits dataset.

**Baselines.** We compared our proposed method with traditional naive baselines and the latest state-of-the-art methods. First, we set the lower bound as the usual supervised sequential fine-tuning result (FT) and the upper bound as the usual supervised joint fine-tuning result. Then, we compared our proposed method with replay-based methods such as ER (Rolnick et al., 2019), Rainbow Memory (RM) (Bang et al., 2021), CLIB (Koh et al., 2021), and CBA (Wang et al., 2023a), regularization-based method LwF (Li & Hoiem, 2017b), SLCA (Zhang et al., 2023), DYSON (He et al., 2024), OnPro (Wei et al., 2023), PEC (Zając et al., 2024) and prompt-based CODA-P (Smith et al., 2023), and MVP (Moon et al., 2023).

**Implementation Details.** We used the MVP (Moon et al., 2023) as our baseline for model and experimental setups. We used Adam optimizer with a learning rate of 5e-3, and implemented with a batch size of 64. As a mapping function $\psi$ which maps prototypes into a relation vector, we used a simple 2-layer MLP function. The hidden dimension of $\psi$ is 64 in CORe50, and 32 in CLEAR100. The size of the dimension of the relation vector $m$ is 10. For the EMA update of global feature topology, a decay factor of 0.99 was used. Experiments were conducted under assumption of the memory-free setting, we adopted naïve reservoir memory for the experiments with memory buffer.

Table 1: Experimental results with proposed Online VIL scenarios. We used bold and underlined as brief indications of the best and the second best, respectively.

| Method | iDigits | | CORe50 | | CLEAR100 | |
|---|---|---|---|---|---|---|
| | $A_{\text{AUC}}$ | $A_{\text{Last}}$ | $A_{\text{AUC}}$ | $A_{\text{Last}}$ | $A_{\text{AUC}}$ | $A_{\text{Last}}$ |
| Upper-bound | - | 87.18±0.13 | - | 91.66±0.25 | - | 94.36±0.28 |
| Lower-bound | 13.45±0.62 | 12.71±3.34 | 3.39±0.10 | 3.24±1.09 | 2.38±0.32 | 2.66±0.55 |
| EWC | 20.07±2.84 | 14.67±3.61 | 18.60±4.64 | 16.07±1.56 | 23.61±3.11 | 19.93±1.33 |
| LwF | 19.61±3.50 | 15.38±1.04 | 25.27±3.77 | 21.97±4.22 | 23.80±2.24 | 21.70±4.19 |
| CODA-P | 23.96±4.74 | 20.62±3.47 | 54.06±5.32 | 48.88±2.92 | 28.82±5.77 | 25.61±3.52 |
| SLCA | 35.81±3.98 | 24.88±2.82 | 33.49±4.47 | 27.36±2.06 | 32.16±2.28 | 31.70±1.13 |
| PEC | 34.77±3.02 | 28.01±2.69 | 51.35±4.39 | 46.95±2.28 | 53.93±3.20 | 51.66±2.09 |
| MVP | 38.29±5.74 | 31.05±3.15 | 58.30±4.48 | 52.84±1.17 | 79.73±3.59 | 77.11±2.33 |
| **TopFlow (Ours)** | **48.52±1.25** | **32.18±1.01** | **64.51 ± 2.50** | **66.20 ± 4.18** | **87.12 ± 0.01** | **80.64 ± 2.67** |

Table 2: Results of proposed OnlineVIL scenarios using replay buffer sizes 500 and 2000.

| Method | Buffer Size=500 | | | | Buffer Size=2000 | | | |
|---|---|---|---|---|---|---|---|---|
| | CORe50 | | CLEAR100 | | CORe50 | | CLEAR100 | |
| | $A_{\text{AUC}}$ | $A_{\text{Last}}$ | $A_{\text{AUC}}$ | $A_{\text{Last}}$ | $A_{\text{AUC}}$ | $A_{\text{Last}}$ | $A_{\text{AUC}}$ | $A_{\text{Last}}$ |
| ER | 74.77±4.85 | 72.25±2.27 | 73.92±3.93 | 71.49±3.20 | 79.44±5.17 | 76.61±3.91 | 83.51±4.61 | 81.03±3.59 |
| RM | 81.06±3.90 | 70.41±3.17 | 72.42±4.64 | 72.93±2.05 | 82.42±3.03 | 79.84±2.19 | 84.73±4.63 | 81.49±2.26 |
| CLIB | 75.06±5.81 | 71.93±1.06 | 68.39±5.25 | 66.92±1.52 | 84.58±4.26 | 81.63±2.51 | 85.62±5.05 | 83.25±1.62 |
| OCM | 75.29±3.10 | 72.66±1.93 | 77.80±3.25 | 75.10±2.42 | 84.92±4.03 | 83.24±2.72 | 84.26±4.82 | 82.91±3.56 |
| CBA | 81.92±4.04 | 81.02±1.39 | 75.26±4.08 | 74.47±1.82 | 85.16±3.28 | 83.49±3.20 | 88.02±3.80 | 85.94±2.36 |
| OnPro | 71.39±4.03 | 70.92±2.24 | 81.36±4.99 | 77.46±1.53 | 81.35±5.51 | 78.09±3.91 | 88.83±5.16 | 86.11±3.57 |
| DYSON | 62.92±5.61 | 60.72±2.16 | 66.56±4.65 | 65.62±2.74 | 51.21±3.71 | 49.29±1.74 | 57.05±4.18 | 55.48±3.43 |
| MVP-R | 83.26±5.16 | 80.16±1.03 | 87.82±3.17 | 85.65±2.18 | 87.33±3.37 | 82.39±1.10 | 89.48±1.71 | 88.93±1.18 |
| **TopFlow (Ours)** | **85.16±0.84** | **91.14±0.05** | **91.67±0.03** | **90.12±1.00** | **87.56±0.82** | **92.24±0.15** | **93.55±0.02** | **92.97±0.65** |

We conducted experiments with 3 random seeds, and note that using more seeds (e.g., 10 runs) also yields consistent results. Details are described in the Appendix Section C.5.1 and Table A.11.

**Evaluation Metrics.** To evaluate online learning performance, we employed two metrics: $A_{\text{AUC}}$ and $A_{\text{Last}}$ (Koh et al., 2021). The $A_{\text{AUC}}$ metric quantifies performance under anytime inference, where inference queries may occur at arbitrary points during training as new classes are encountered. Conversely, $A_{\text{Last}}$ assesses inference accuracy after training. In real-world applications, models must deliver reliable predictions on demand, regardless of training stage. Thus, $A_{\text{AUC}}$ and $A_{\text{Last}}$ provide a robust framework for benchmarking online learning performance in practical deployment settings.

## 4.2 EXPERIMENTAL RESULTS

We conducted extensive experiments in the proposed Online VIL scenario, and the results are summarized in Table 1 and Table 2. As shown in the Table 1, non-replay setting, our proposed TopFlow consistently outperforms existing methods in terms of both $A_{\text{AUC}}$ and $A_{\text{Last}}$. These results demonstrate that our approach mitigates catastrophic forgetting while enabling continual adaptation to the input stream.

Moreover, in Table 2, the introduction of replay memory generally improves performance, but the proposed method also consistently outperforms all the baselines. Surprisingly, in replay-buffer settings, we observe that naïve methods such as Experience Replay (ER) and earlier methods tend to achieve the best performance, except for our proposed method. This suggests that most existing online-incremental learning methods struggle in realistic Online VIL scenarios where distribution shifts are frequent and task boundaries are unclear. In contrast, our proposed TopFlow maintains strong performance across all settings, confirming its robustness.

Furthermore, TopFlow achieves a more stable accuracy trajectory over time, with fewer drastic performance drops between tasks. This indicates that DFM and GTP help smooth the learning process by leveraging more structured representations and effective feature alignment.

## 4.3 ABLATION STUDIES AND ANALYSIS

**Ablation Study.** Table 3 demonstrates the effectiveness of each component in our TopFlow framework. Implementing DFM or GTP individually significantly enhanced both $A_{\text{AUC}}$ and $A_{\text{Last}}$, validating their respective contributions. DFM stabilizes learning by distilling domain-invariance self-knowledge while creating representational capacity for class discrimination in posterior layers. GTP ensures steady performance regardless of batch composition while particularly boosting final accuracy through

Table 3: Ablation study for DFM and GTP on CORe50.

| DFM | GTP | $A_{\text{AUC}}$ | $A_{\text{Last}}$ |
|---|---|---|---|
| Baseline | | 58.30 | 52.84 |
| ✓ | | 64.13 | 64.57 |
| | ✓ | 63.95 | 65.28 |
| ✓ | ✓ | **64.51** | **66.20** |

Table 4: The Effectiveness of TopFlow in Standard CL Scenarios.

| Method | Accuracy | |
|---|---|---|
| | CIL, CODA-P | DIL, S-Prompt |
| Baseline | 84.17 | 82.96 |
| + DFM | 84.19 | 85.36 |
| + GTP | 85.52 | 86.88 |
| + DFM, GTP | **86.04** | **87.50** |

Table 5: Ablation of layer selection of DFM loss.

| Layer | $A_{\text{Last}}$ |
|---|---|
| Baseline | 52.19 |
| (0, 5) | 49.92 |
| (5, 10) | 52.98 |
| (5, 10), (6, 11) | 53.49 |
| **(6, 11) (Ours)** | **54.82** |

its ability to integrate transient batch-specific insights into persistent feature topology. Combining these methods yields complementary results, with improved representations fed into posterior layers by DFM working in concert with the topological knowledge preservation of GTP to achieve superior performance in the challenging Online VIL scenario.

**Effectiveness of TopFlow in Standard CL Scenarios.** As shown in the Table 4, DFM and GTP improve performance in both Standard CIL and DIL, individually and combined. We adopted the widely used CIFAR-100 (Krizhevsky et al., 2009) 10-split CIL scenario and CORe50 (Lomonaco & Maltoni, 2017) 8-split DIL scenario as standards, and conducted experiments with CODA-P (Smith et al., 2023) and S-Prompts (Wang et al., 2022a) as baselines for CIL and DIL respectively. Despite the limited domain variation in CIL, the proposed DFM and GTP showed performance improvements. In DIL, both methods yield significant improvements, highlighting their intense motivation and effectiveness under real-world domain shifts.

Table 6: Performance comparison of different methods and variants.

| Method | $A_{\text{Last}}$ |
|---|---|
| CODA-P | 49.13 |
| + DFM | 52.88 |
| + GTP | 53.17 |
| **+ DFM, GTP** | **54.62** |
| PEC | 45.99 |
| + DFM | 46.64 |
| + GTP | 48.19 |
| **+ DFM, GTP** | **48.86** |

**Layer Selection for DFM.** To analyze the effect of layer selection for our proposed DFM loss, we conduct an ablation study using various $(n, l)$ pairs for Equation 5 from the ViT encoder. As shown in Table 5, applying DFM loss on early layers such as (0, 5) leads to a performance drop compared to the baseline, suggesting that early-layer features are not well-aligned with the high-level semantics captured in the final layer. On the other hand, using higher layers such as (5, 10) or a combination like (5, 10), (6, 11) yields improved performance, with the latter reaching 53.49. This indicates that deeper layers better preserve semantic information beneficial for matching with final representations. Our final configuration, using layers (6, 11) for DFM loss, achieves the best performance with 54.82. This choice strikes a balance between representational abstraction and compatibility with final-layer features, enhancing the stability and effectiveness of the DFM loss.

**DFM and GTP with Other Models.** As a generally designed framework, we consider DFM and GTP not only to be effective independently but also to generate synergy when integrated. We conducted ablation experiments to verify whether the proposed DFM and GTP could also yield performance improvements on baseline CL algorithms other than the MVP. As shown in Table 6, DFM and GTP lead to performance improvements over the baseline, even independently. DFM and GTP robustly improve performance over the baseline on the CORe50 dataset and other existing continual learning algorithms used in the main experiments.

## 5 CONCLUSION

We proposed a new online continual learning scenario named Online VIL, which simulates a complex real world where states are ever-changing and there are no concepts of tasks and clear boundaries between them. Through analysis, we determined the direction for problem-solving in Online VIL and defined novel TopFlow framework. We demonstrated that the proposed TopFlow showed SOTA performance in the challenging Online VIL scenario, and its effectiveness through various experiments. Online VIL involves stochastic task construction, where the composition of data to each task is influenced by random seed. While this design captures more realistic dynamics, it can introduce variability in results. Despite this, we hope that our Online VIL scenario will serve as a new benchmark for advancing real-world incremental learning research, providing a more realistic and challenging setting for future studies.

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

APPENDIX

In this Appendix, we provide additional details and further analysis on the **Online VIL** scenario and proposed **TopFlow** as follows:

# A  ALGORITHMS AND DETAILS FOR METHODS

## A.1  ONLINE VIL

Algorithm A.1 shows the actual configuration procedure for an Online VIL scenario configuration. The RandSplit$(S, C)$ function partitions a set of items $S$ into $C$ chunks. Given a set of samples $S$ and a target number of chunks $C$, RandSplit first creates a random permutation of the items in $S$. If $C > 1$, it randomly selects $C - 1$ distinct division points from the $|S| - 1$ possible positions between the permuted items. These points partition the permuted sequence into $C$ chunks, which may thus have variable sizes. The function returns an ordered list of these $C$ disjoint chunks, whose union is $S$. If $C = 1$, the single chunk returned is $S$. The Sample$(X, k)$ function, when $X$ is a set or collection, randomly selects $k$ distinct items from $X$ without replacement. If $X$ is a category identifier, Sample$(X, k)$ implies sampling from the underlying data instances associated with category $X$. If $X$ is a range like $\{1, ..., M\}$, it samples an integer uniformly from that range.

In Figure A.1, we demonstrate an example of the density of these created Tasks over time. It can be seen that it is possible to dynamically control the unbalanced, variable, online learning coverage induced by the task configuration. In our experiments, we found that beta acts like a memory to blur task boundaries when the domain changes significantly. Since utilizing direct task boundaries due to the online batch is impossible, we wanted to check the model's performance more objectively by not applying the task boundary blurring beta.

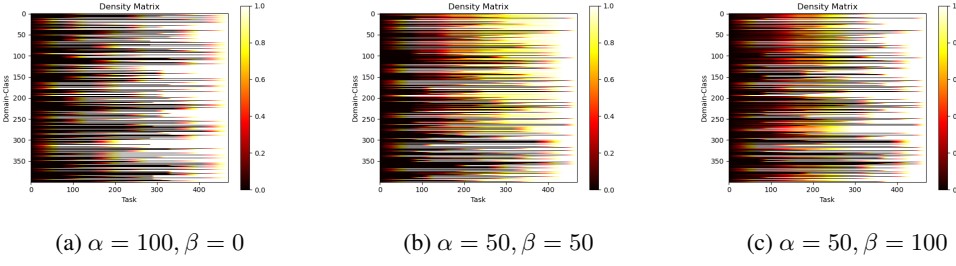

(a) $\alpha = 100, \beta = 0$        (b) $\alpha = 50, \beta = 50$        (c) $\alpha = 50, \beta = 100$

Figure A.1: Visualization of ratio of sample visibility under various $\alpha, \beta$

---

**Algorithm A.1** Online VIL Task Construction

---

1: **Input:** Categories $\mathcal{K}$, disjoint ratio $\alpha$, blurry ratio $\beta$, tasks $M$, batch size $b$
2: **Output:** Batch sequence $\mathcal{B} = \{B_1, B_2, ..., B_N\}$

3:   /* category partitioning */
4: $\mathcal{K}_{\text{disjoint}} \leftarrow \text{Sample}(\mathcal{K}, \lfloor \alpha|\mathcal{K}| \rfloor)$                 $\triangleright$ Categories with clear boundaries
5: $\mathcal{K}_{\text{blurry}} \leftarrow \mathcal{K} \setminus \mathcal{K}_{\text{disjoint}}$                  $\triangleright$ Categories with ambiguous boundaries

6:   /* sample extraction */
7: **for** each $k \in \mathcal{K}_{\text{blurry}}$ **do**
8:     $\mathcal{S}_k^{\text{blurred}} \leftarrow \text{Sample}(k, \beta|k|)$              $\triangleright$ Samples across the boundaries
9:     $\mathcal{S}_k^{\text{nonblurred}} \leftarrow k \setminus \mathcal{S}_k^{\text{blurred}}$            $\triangleright$ Samples remain in the boundaries
10: **end for**

11:   /* sample redistribution */
12: $\mathcal{K}_{\text{blurred}} \leftarrow \text{RandSplit}\left( \bigcup_{k \in \mathcal{K}_{\text{blurry}}} \mathcal{S}_k^{\text{blurred}}, |\mathcal{K}_{\text{blurry}}| \right)$

13: $\mathcal{K}_{\text{nonblurred}} \leftarrow \{ \mathcal{S}_k^{\text{nonblurred}} \mid k \in \mathcal{K}_{\text{blurry}} \}$

14:   /* random task assignment */
15: **for** each category $i$ in $\mathcal{K}$ **do**
16:     $t_i \sim \text{Uniform}\{1, ..., M\}$
17: **end for**

18:   /* task construction */
19: **for** $k = 1$ to $M$ **do**
20:     $T_k^{\text{disjoint}} \leftarrow \{i \in \mathcal{K}_{\text{disjoint}} \mid t_i = k\}$
21:     $T_k^{\text{nonblurred}} \leftarrow \{i \in \mathcal{K}_{\text{nonblurred}} \mid t_i = k\}$
22:     $T_k^{\text{blurred}} \leftarrow \{i \in \mathcal{K}_{\text{blurred}} \mid t_i = k\}$
23:     $T_k \leftarrow T_k^{\text{disjoint}} \cup T_k^{\text{nonblurred}} \cup T_k^{\text{blurred}}$     $\triangleright$ Task $k$ with explicit and implicit boundary
24: **end for**

25:   /* batch generation */
26: $\mathcal{B} \leftarrow \{\}$
27: **for** $k = 1$ to $N$ **do**
28:     **while** $\|\mathcal{T}_k\| > 0$ **do**
29:         $B_t \leftarrow \text{Sample}(T_k, \min(b, \|T_k\|))$
30:         $\mathcal{B} \leftarrow \mathcal{B} \cup B_t$
31:         $T_k \leftarrow T_k \setminus B_t$
32:     **end while**
33: **end for**
34: **return** $\mathcal{B}$                            $\triangleright$ Online VIL Batch Sequence

## A.2 DOMAIN-AGNOSTIC FLOW MATCHING

The inner product between two continuous real functions in the same function space $f, g \in C : X \mapsto Y, x \in X \subset \mathbb{R}^{d_x}, y \in Y \subset \mathbb{R}^{d_y}$ is defined with a Riemannian integral:

$$\langle f(x), g(x) \rangle = \int_X f(x)^\top g(x) d\boldsymbol{x} \tag{A.1}$$

With an open and neighborhood set $\mathcal{V}_t$ is a small subset of the domain $X$, the feature $\boldsymbol{h}_n$ from the $n$-th layer can be expressed with infinitesimal variation on the tangent space $d\boldsymbol{h}_{n-1}$:

$$d\boldsymbol{h}_{n+1} = (D_{\boldsymbol{h}_n} f_{n+1} + I) \, d\boldsymbol{h}_n, \quad d\boldsymbol{h}_l = \left( \prod_{k=n+1}^l \left( D_{\boldsymbol{h}_{k-1}} f_k + I \right) \right) d\boldsymbol{h}_n = \Phi_n^l d\boldsymbol{h}_n, \tag{A.2}$$

where $D_{\boldsymbol{h}_n} f_{n+1}$ is the Fréchet derivative of $f_{n+1}$ at $\boldsymbol{h}_n$, and $I$ is an identity matrix, and $\Phi_n^l$ is a push-forward map from tangent space $TF_n \mapsto TF_l$.

The covariance matrix for left singular matrix $U$ is:

$$\frac{1}{b-1} H^\top H = \left( U\Sigma V^\top \right)^\top U\Sigma V = V\Sigma^\top U^\top U\Sigma V^\top = V\Sigma^\top \Sigma V^\top$$

$$= \mathbb{E}_{\boldsymbol{x} \in \mathcal{V}_t} \left[ \begin{bmatrix} \boldsymbol{h}_n^\top & \boldsymbol{h}_l^\top \end{bmatrix} \begin{bmatrix} \boldsymbol{h}_n \\ \boldsymbol{h}_l \end{bmatrix} \right]$$

$$= \mathbb{E}_{\boldsymbol{x} \in \mathcal{V}_t} \left[ \bar{\boldsymbol{h}}_n^\top \bar{\boldsymbol{h}}_n + \bar{\boldsymbol{h}}_n^\top \bar{\boldsymbol{h}}_l + \bar{\boldsymbol{h}}_l^\top \bar{\boldsymbol{h}}_n + \bar{\boldsymbol{h}}_l^\top \bar{\boldsymbol{h}}_l + d\boldsymbol{h}_n^\top d\boldsymbol{h}_n + d\boldsymbol{h}_n^\top d\boldsymbol{h}_l + d\boldsymbol{h}_l^\top d\boldsymbol{h}_n + d\boldsymbol{h}_l^\top d\boldsymbol{h}_l \right],$$

$$= \mathbb{E}_{\boldsymbol{x} \in \mathcal{V}_t} \left[ \bar{\boldsymbol{h}}_n^\top \bar{\boldsymbol{h}}_n + d\boldsymbol{h}_n^\top d\boldsymbol{h}_n \right] + \mathbb{E}_{\boldsymbol{x} \in \mathcal{V}_t} \left[ d\boldsymbol{h}_l^\top d\boldsymbol{h}_l + \bar{\boldsymbol{h}}_l^\top \bar{\boldsymbol{h}}_l \right]$$

$$+ \mathbb{E}_{\boldsymbol{x} \in \mathcal{V}_t} \left[ \bar{\boldsymbol{h}}_n^\top \bar{\boldsymbol{h}}_l + \bar{\boldsymbol{h}}_l^\top \bar{\boldsymbol{h}}_n + d\boldsymbol{h}_n^\top d\boldsymbol{h}_l + d\boldsymbol{h}_l^\top d\boldsymbol{h}_n \right],$$

$$\simeq \langle \boldsymbol{h}_n, \boldsymbol{h}_n \rangle + \langle \boldsymbol{h}_l, \boldsymbol{h}_l \rangle + \langle \boldsymbol{h}_n, \boldsymbol{h}_l \rangle, \tag{A.3}$$

where $\langle \bar{\boldsymbol{h}}_i, d\boldsymbol{h}_i \rangle \simeq 0$ for any $i$-th layer. The last layer feature $\boldsymbol{h}_l$ can be denoted with the variation of intermediate layer $\boldsymbol{h}_n$ as:

$$\boldsymbol{h}_l = \bar{\boldsymbol{h}}_l + d\boldsymbol{h}_l = \bar{\boldsymbol{h}}_l + D_{\boldsymbol{h}_n} \boldsymbol{h}_l d\boldsymbol{h}_n. \tag{A.4}$$

The third term in Equation A.3 can be expressed as:

$$\langle \boldsymbol{h}_n, \boldsymbol{h}_n \rangle + \langle \boldsymbol{h}_l, \boldsymbol{h}_l \rangle + \langle \boldsymbol{h}_n, \boldsymbol{h}_l \rangle$$

$$\simeq \bar{\boldsymbol{h}}_n^\top \bar{\boldsymbol{h}}_n + \bar{\boldsymbol{h}}_n^\top \bar{\boldsymbol{h}}_l + \bar{\boldsymbol{h}}_l^\top \bar{\boldsymbol{h}}_n + \bar{\boldsymbol{h}}_l^\top \bar{\boldsymbol{h}}_l$$

$$+ \mathbb{E}_{\boldsymbol{x} \in \mathcal{V}_t} \left[ d\boldsymbol{h}_n^\top d\boldsymbol{h}_n + d\boldsymbol{h}_n^\top d\boldsymbol{h}_l + d\boldsymbol{h}_l^\top d\boldsymbol{h}_n + d\boldsymbol{h}_l^\top d\boldsymbol{h}_l \right]$$

$$= \bar{\boldsymbol{h}}_n^\top \bar{\boldsymbol{h}}_n + \bar{\boldsymbol{h}}_n^\top \bar{\boldsymbol{h}}_l + \bar{\boldsymbol{h}}_l^\top \bar{\boldsymbol{h}}_n + \bar{\boldsymbol{h}}_l^\top \bar{\boldsymbol{h}}_l$$

$$+ \mathbb{E}_{\boldsymbol{x} \in \mathcal{V}_t} \left[ d\boldsymbol{h}_n^\top d\boldsymbol{h}_n + d\boldsymbol{h}_n^\top \Phi_n^l \delta\boldsymbol{h}_n + d\boldsymbol{h}_n^\top \left( \Phi_n^l \right)^\top d\boldsymbol{h}_n + d\boldsymbol{h}_n^\top \left( \Phi_n^l \right)^\top \Phi_n^l d\boldsymbol{h}_n \right]. \tag{A.5}$$

In here, the term of $\mathbb{E}_{\boldsymbol{x} \in \mathcal{V}_t} \left[ d\boldsymbol{h}_n^\top \Phi_n^l \delta\boldsymbol{h}_n + d\boldsymbol{h}_n^\top \left( \Phi_n^l \right)^\top d\boldsymbol{h}_n \right]$ denotes the common space between $\boldsymbol{h}_n$ and transformed by the push-forward, and the term of $\mathbb{E}_{\boldsymbol{x} \in \mathcal{V}_t} \left[ d\boldsymbol{h}_n^\top \left( \Phi_n^l \right)^\top \Phi_n^l d\boldsymbol{h}_n \right]$ provides a information about the push-forward metric induced by $\boldsymbol{h}_n$. Because $U^\top U = I$, we can use the $U^\top$ as a projection onto the common space. Also a pertubated function $\boldsymbol{h}_n^*$ from $\boldsymbol{h}_n$ can be projected onto common space, the DFM loss in Equation 5 optimize both of the diagonal terms $\langle \boldsymbol{h}_n, \boldsymbol{h}_n \rangle$, $\langle \boldsymbol{h}_l, \boldsymbol{h}_l \rangle$ and cross-layer interaction $\langle \boldsymbol{h}_n, \boldsymbol{h}_l \rangle$ in cosine similarity and the push-forward metric of local neighborhood of sample point, modify the local geometric structure.

## A.3 GLOBAL TOPOLOGY PRESERVATION

The overall algorithm for Global Topology Preservation is indicated in Algorithm A.3. For each iteration, $n_k$ prototypes are generated with the first hierarchy level of clustering of FINCH clustering. We used the Frobenius norm as a cost function for the Hungarian algorithm.

# B EXPERIMENTAL DETAILS

## B.1 DATASETS

We conducted experiments on three benchmark datasets, iDigits (Volpi et al., 2021), CORe50 (Lomonaco & Maltoni, 2017), and CLEAR100 (Lin et al., 2021), which are all suitable for constructing incremental learning scenarios characterized by substantial distribution shifts across both class semantics and visual domains. These benchmarks were selected for their ability to clearly distinguish

---

**Algorithm A.2** Domain-agnostic Flow Matching (DFM) Loss Computation

---

1: **Input:** Batch of samples $\boldsymbol{X}$, frozen intermediate layer $f_i$, frozen last layer $f$, prompted intermediate layer $f_i^*$, subspace dimension $k$, temperature $\tau$.
2: **Output:** Domain-agnostic Flow Matching Loss $\mathcal{L}_{\text{DFM}}$.

3:   /* feature extraction */
4: $\boldsymbol{P}_{V^*} \leftarrow f_i^*(\boldsymbol{x})$ for $\boldsymbol{x} \in \boldsymbol{X}$                    ▷ Prompted intermediate features
5: $\boldsymbol{P}_{VL} \leftarrow f(\boldsymbol{x})$ for $\boldsymbol{x} \in \boldsymbol{X}$                      ▷ Frozen last-layer features
6: $\boldsymbol{P}_V \leftarrow f_i(\boldsymbol{x})$ for $\boldsymbol{x} \in \boldsymbol{X}$                     ▷ Frozen intermediate features

7:   /* geodesic flow kernel (GFK) computation */
8: $\boldsymbol{G}_{V^* \leftrightarrow V} \leftarrow \text{GFK}(\boldsymbol{P}_{V^*}, \text{concat}(\boldsymbol{P}_{V^L}, \boldsymbol{P}_V))$

9:   /* DFM loss calculation */
10: $\mathcal{L}_{\text{DFM}} \leftarrow 0$
11: $B \leftarrow \text{BatchSize}(\boldsymbol{X})$
12: **for** $m = 1$ to $B$ **do**
13:     $\texttt{anchor} \leftarrow \boldsymbol{F}_{v^*}[m,:]$
14:     $\texttt{positive} \leftarrow \boldsymbol{F}_{vL}[m,:]$
15:     $S_{pos} \leftarrow \text{Similarity}(\texttt{anchor}, \texttt{positive}, \boldsymbol{G}_{V^* \leftrightarrow V})$
16:     $\texttt{numerator\_sum} \leftarrow \exp(S_{pos}/\tau)$
17:     $\texttt{denominator\_sum} \leftarrow 0$
18:     **for** $n = 1$ to $B$ **do**
19:         **if** $n \neq m$ **then**
20:             $\texttt{neg}_{\text{L}} \leftarrow \boldsymbol{F}_{vL}[n,:]$
21:             $S_{\text{neg\_L}} \leftarrow \text{Similarity}(\texttt{anchor}, \texttt{neg}_{\text{L}}, \boldsymbol{G}_{V^* \leftrightarrow V})$
22:             $\texttt{denominator\_sum} \leftarrow \texttt{denominator\_sum} + \exp(S_{\text{neg\_L}}/\tau)$
23:         **end if**
24:         $\texttt{neg}_{\text{I}} \leftarrow \boldsymbol{F}_v[n,:]$          ▷ Frozen intermediate features as negatives
25:         $S_{\text{neg\_I}} \leftarrow \text{Similarity}(\texttt{anchor}, \texttt{neg}_{\text{I}}, \boldsymbol{G}_{V^* \leftrightarrow V})$
26:         $\texttt{denominator\_sum} \leftarrow \texttt{denominator\_sum} + \exp(S_{\text{neg\_I}}/\tau)$
27:     **end for**
28:     $\mathcal{L}_{\text{DFM}} \leftarrow \mathcal{L}_{\text{DFM}} - \log(\texttt{numerator\_sum}/\texttt{denominator\_sum})$
29: **end for**
30: $\mathcal{L}_{\text{DFM}} \leftarrow \mathcal{L}_{\text{DFM}}/B$
31: **return** $\mathcal{L}_{\text{DFM}}$

---

between class and domain changes, enabling the rigorous evaluation of the Online VIL scenario. The composition of each dataset is summarized in Table A.1.

For iDigits, we follow the incremental scenario introduced in (Volpi et al., 2021), which comprises four digit datadsets: MNIST (LeCun et al., 1998), SVHN (Netzer et al., 2011), MNIST-M, and SynDigits (Ganin & Lempitsky, 2015). Each dataset is regarded as a distinct domain while sharing the same digit class space, making it an ideal setup for evaluating the interplay of domain and class shift in a controlled setting.

CORe50 is a widely used benchmark for domain-incremental learning in real-world object recognition. It contains 50 household object classes captured under 11 domain conditions that vary in background, lighting, and acquisition settings. In our experimental setup, we use samples from 8 domains for training and hold out the remaining three as unseen test domains. This allows us to assess the generalization ability of the model to novel domain variations encountered post-training.

CLEAR100 is another standard benchmark for continual and domain-incremental learning, comprising 100 object categories collected from 10 visually diverse domains. In contrast to other benchmarks such as DomainNet (Peng et al., 2019), CLEAR100 is carefully constructed with a temporally ordered domain progression, where domain shifts occur in a sequence aligned with a chronological timeline. This design mimics real-world scenarios where environments evolve gradually, allowing more realistic evaluation of online learning systems in terms of their adaptability, robustness, and resistance to catastrophic forgetting. Additionally, the high domain diversity in CLEAR100 makes it particularly challenging and relevant for studying dynamic representation learning.

---

**Algorithm A.3** Global Topology Preservation (GTP) for Online VIL

---

1: **Input:** Batch $B_t$, Backbone with prompt tuning $f_p(\cdot)$, Global prototypes $\bar{P}_g = \{\bar{p}_g^j\}_{j=1}^k$, EMA rate $\alpha$, Mapping function $\psi(\cdot)$, Distance metric $D(\cdot, \cdot)$, Finch clustering algorithm $\text{FINCH}(\cdot)$.

2: **Output:** GTP loss $\mathcal{L}_{\text{GTP}}$

3:    /* clustering */

4: $\text{k} \leftarrow \|\bar{P}_g\|$                                                     ▷ Number of global prototypes

5: $P_b = \{p_b^j\}_{j=1}^{n_k} \leftarrow \text{FINCH}(f_p(B_t))$             ▷ $n_k$ clusters with Finch clustering

6:    /* Hungarian algorithm */

7: **if** $n_k > k$ **then**

8:     $\bar{P}_g^{\text{dummy}} \leftarrow \{\mathbf{0}, \mathbf{0}, \ldots\}$ with size $(n_k - k)$

9:     $\bar{P}_g \leftarrow \bar{P}_g \cup \bar{P}_g^{\text{dummy}}$

10: **end if**

11: $S_{n_k} \leftarrow \{\pi : \{1, \ldots, n_k\} \rightarrow \{1, \ldots, n_k\} \mid \pi \text{ is bijection}\}$

12: $\pi^* = \arg\min_{\pi \in S_{n_k}} \sum_{i=1}^{n_k} \|p_b^i - \bar{p}_g^{\pi(i)}\|_F, \; p_b^i \in P_b, \; \bar{p}_g^{\pi(i)} \in \bar{P}_g$    ▷ Hungarian algorithm

13:    /* global prototype update */

14: **for** $i = 1$ to $n_k$ **do**

15:     **if** $\pi^*(i) \leq k$ **then**

16:         $\bar{p}_g^{\pi^*(i)} \leftarrow (1 - \alpha)\bar{p}_g^{\pi^*(i)} + \alpha p_b^i$          ▷ EMA update global prototypes

17:     **else**

18:         $p_g^i \leftarrow p_b^i$

19:     **end if**

20: **end for**

21:    /* GTP loss calculation */

22: $\mathcal{L}_{\text{GTP}} \leftarrow 0$                                                  ▷ Initialize GTP loss

23: **for** $i = 1$ to $n_k$ **do**

24:     **for** $j = 1$ to $n_k$ **do**

25:         **if** $\pi^*(i) \neq \pi^*(j)$ **then**

26:             $\mathcal{L}_{\text{GTP}} \leftarrow \mathcal{L}_{\text{GTP}} + D\big(\psi(p_b^i, p_b^j), \psi(\bar{p}_g^{\pi^*(i)}, \bar{p}_g^{\pi^*(j)})\big)$

27:         **end if**

28:     **end for**

29: **end for**

30: **return** $\mathcal{L}_{\text{GTP}}$

---

Table A.1: Dataset composition used in experiments.

| Dataset | Composition | |
| --- | --- | --- |
| | #Class | #Domain |
| iDigits Volpi et al. (2021) | 10 | 4 |
| CORe50 Lomonaco & Maltoni (2017) | 50 | 11 |
| CLEAR100 (Lin et al., 2021) | 100 | 10 |

## B.2 IMPLEMENTATION DETAILS.

We conducted all the experiments on a single NVIDIA GeForce RTX 3090 GPU and 8 Intel(R) Xeon(R) Gold 6226R CPU cores. To make fair comparisons, we used standard ImageNet Deng et al. (2009) pre-trained ViT-B/16[1] Dosovitskiy et al. (2020) as a backbone of all methods. We used MVP (Moon et al., 2023) as a baseline of our architecture and added the proposed $\mathcal{L}_{\text{DFM}}$ and $\mathcal{L}_{\text{GTP}}$ along with the loss of MVP. Almost all online incremental learning baselines assumed multi-iteration training (Aljundi et al., 2019), so we also adopted 3 times augmented iteration comparisons with our proposed method and other baselines.

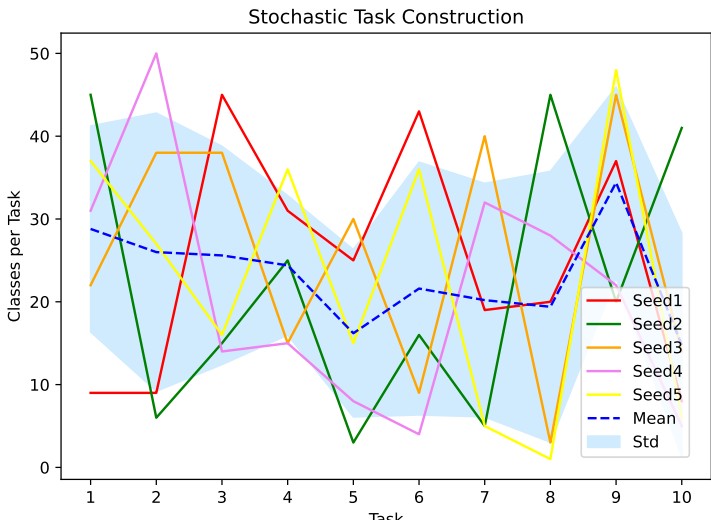

Figure A.2: Classes per task constructed from different seeds.

### B.3 NUMBER OF CLASSES IN THE TASK

As mentioned in Algorithm A.1, we used random selection for task construction. The task construction process, where a random seed randomly determines the class composition and number of classes per task under a fixed total class budget, naturally reflects the stochasticity and variability of real-world data streams. This setting departs from the conventional incremental learning benchmarks, which typically assume a fixed and balanced class allocation per task. Instead, our design introduces variability in task granularity and uncertainty in class arrival patterns, two key properties of naturalistic data distributions.

Figure A.2 visualizes the distribution of class counts per task over 10 incremental steps across five random seeds on the CORe50 dataset. Each colored line represents a different seed (Seed1–Seed5), with the dashed blue line showing the mean trend and the shaded region indicating one standard deviation. Notably, the number of classes per task varies significantly within and across seeds, ranging from as few as two classes to over 50. This variability closely reflects real-world deployment scenarios in which task boundaries are ill-defined and concept drift occurs with unpredictable granularity and cadence.

## C ADDITIONAL EXPERIMENTS AND ANALYSIS

### C.1 OTHER DATASETS

**TinyImageNet.** We extended our evaluation to TinyImageNet (200 classes, 100K images) dataset, which presents greater visual complexity and a larger label space than the datasets used in the main paper. The results in Table A.2 demonstrate that TopFlow maintains its performance advantage on TinyImageNet, indicating its ability to handle more challenging, real-world-like scenarios while generalizing beyond small-scale benchmarks. The consistent improvement validates our method's scalability to larger, more complex datasets.

Table A.2: Evaluation on TinyImageNet.

| Method | $A_{\text{AUC}}$ | $A_{\text{Last}}$ |
|---|---|---|
| MVP | 52.13 | 30.51 |
| **TopFlow (Ours)** | **52.39** | **33.74** |

---

[1]storage.googleapis.com/vit_models/imagenet21k/ViT-B_16.npz

**UCF-101.** We conducted experiments on UCF-101, a challenging video action recognition dataset containing 101 human action classes from videos captured in the wild. It presents significant challenges, including varying lighting conditions, camera angles, background clutter, and temporal dynamics—properties that closely mirror real-world deployment scenarios. Our proposed TopFlow demonstrates consistent improvements even in these more challenging, real-world-like environments, as demonstrated in Table A.3. This validates the ability of the TopFlow to handle the temporal dynamics and visual complexity present in genuinely "wild" data streams, addressing your concern about evaluation beyond curated academic datasets.

Table A.3: Evaluation on UCF-101.

| Method | $A_{\text{AUC}}$ | $A_{\text{Last}}$ |
|---|---|---|
| MVP | 57.21 | 33.03 |
| **TopFlow (Ours)** | **57.42** | **37.84** |

## C.2 VARIATION ON MODEL COMPONENTS

### C.2.1 MODEL BACKBONE

**Model Random Initialization.** For further analysis, we conducted experiments comparing the baseline and TopFlow using randomly initialized backbones. The results in Table A.4 indicate that our performance improvements are not dependent on pre-training and remain effective when the model is trained from scratch. This reveals that the proposed DFM and GTP components perform as intended, demonstrating their intrinsic effectiveness rather than merely relying on pretrained knowledge. The consistent improvement of pretrained and random initialization settings confirms that our gains stem from methodological innovations.

Table A.4: Evaluation on randomly initialized backbone.

| Method | $A_{\text{AUC}}$ | $A_{\text{Last}}$ |
|---|---|---|
| MVP | 14.65 | 4.50 |
| **TopFlow (Ours)** | **16.04** | **5.72** |

### C.2.2 DIMENSION FOR DFM LOSS

**Dimension Size in DFM.** We conducted experiments comparing explicit dimension selection with our implicit approach. The results in Table A.5 demonstrate that our implicit contrastive approach outperforms explicit fixed-size identification, validating the theoretical motivation while providing a practically superior implementation. These results prove the superiority of the proposed DFM, which was derived through a well-connected method of thorough theoretical analysis, recognition of related difficulties, and introduction of countermeasures.

Table A.5: Performance across different dimension sizes.

| Dimension Size | $A_{\text{Last}}$ |
|---|---|
| Baseline | 52.19 |
| 4 | 53.22 |
| 16 | 53.17 |
| 64 | 53.90 |
| 128 | 53.88 |
| 512 | 54.26 |
| **768 (Ours)** | **54.82** |

## C.3 ROBUSTNESS ON VARIOUS SCENARIOS

**GTP with Few Unique Classes.** While the goal of Online VIL is continuous and broad variation of the scenario, each batch can contain only a few unique classes in Online VIL, which may affect

the clustering of GTP module. Hence, we conducted experiments using 3 randomly selected seeds, each containing 3 sessions (10 sessions total) with a few unique classes (average 2.3 classes, 50 classes) from the CORe50 dataset. We observed stable performance, even with a few unique classes, as indicated in Table A.8. GTP maintains stable performance even under sparse class conditions, demonstrating robustness to the unpredictable streams characteristic of Online VIL.

Table A.6: Comparison of results across different scenarios.

| Scenario | $A_{\textbf{Last}}$ |
|---|---|
| Main Table Results | $66.20 \pm 4.18$ |
| Results with Few Unique Classes | $65.93 \pm 4.34$ |

**Comparision with (Park et al., 2024b)** We a evaluate TopFlow and ICON in Standard VIL and Online VIL scenario. ICON relies on assumptions of discrete task changes that no longer apply in the online scenario. In contrast, TopFlow maintains robust performance without needing task-level information or multiple passes.

Table A.7: Comparison with ICON Park et al. (2024b) on Virsatil Incremental Learining scinario.

| Scenario | $A_{\textbf{Last}}$ |
|---|---|
| ICON | $45.15 \pm 2.94$ |
| TopFlow (Ours) | $66.20 \pm 4.18$ |

Table A.8: Comparison with ICON on Online VIL scinario.

| Scenario | $A_{\textbf{Last}}$ |
|---|---|
| ICON | $79.06 \pm 3.43$ |
| TopFlow (Ours) | $81.20 \pm 2.75$ |

### C.3.1 CLUSTERING METHOD FOR GTP

Table A.9 demonstrates clustering method effects on GTP. K-Means (Hartigan & Wong, 1979) shows fast adaptation but limited generalization due to its Euclidean geometry assumption. DBSCAN (Ester et al., 1996) outperforms K-Means through better graph-spectral geometry estimation, while FINCH (Sarfraz et al., 2019) achieves superior results via parameter-free hierarchical clustering that adapts to variable batch distributions. The significant performance gap between K-Means and topology-aware methods (DBSCAN and FINCH) confirms that the feature space contains substantial nonlinear relationships. This validates our Domain-agnostic Flow Matching approach, which accumulates structural information in earlier layers while preserving model expressivity.

Table A.9: Analysis of Clustering Methods for GTP.

| Method | $A_{\text{AUC}}$ | $A_{\text{Last}}$ |
|---|---|---|
| Baseline | 58.30 | 52.84 |
| K-Means | 64.04 | 60.44 |
| DBSCAN | 64.46 | 66.18 |
| **FINCH** | **64.51** | **66.20** |

### C.4 VISUALIZATION OF GTP

To qualitatively analyze the effect of GTP, we visualize the output features using t-SNE (van der Maaten & Hinton, 2008) over tasks with and without GTP in Figure A.3 and A.4, respectively. We track the evolution of features from the first 10 classes, which are available for all tasks in the CORe50 dataset. And low-dimensional approximation is performed across all tasks to maintain consistency and relative positions of features. While it is challenging to project high-dimensional topological structures into 2D space, we observe that GTP helps to preserve the relative arrangement and internal

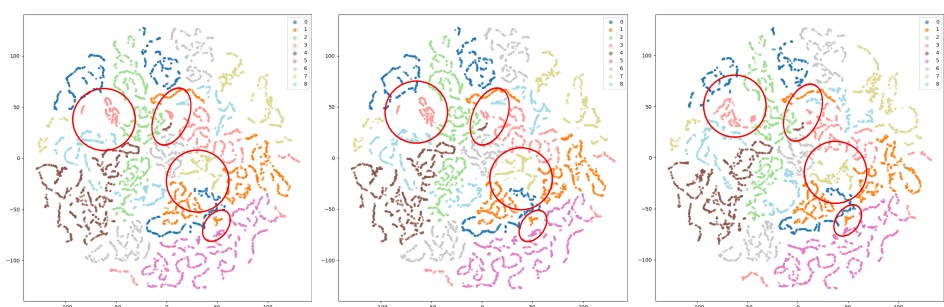

Figure A.3: t-SNE visualization of output features over tasks with GTP.

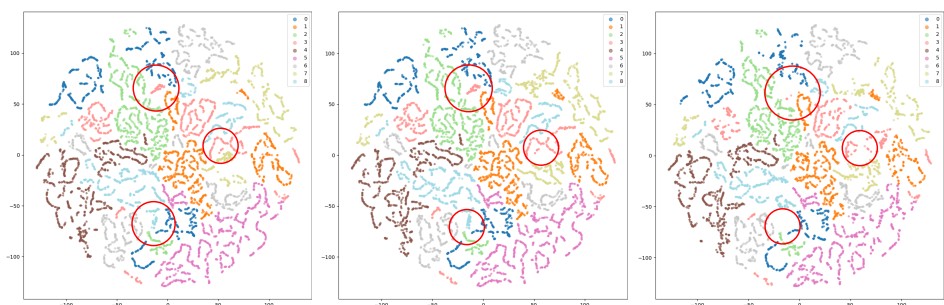

Figure A.4: t-SNE visualization of output features over tasks without GTP.

structure of clusters across tasks. Especially in red circles, the meaningful topologies (e.g., relative connections or penetrations between clusters) are better maintained with GTP, whereas they drift and dismorph without it. This qualitative analysis supports our quantitative findings, demonstrating that GTP effectively preserves global topological relationships in the feature space during online learning.

## C.5 HYPERPARAMETERS

### C.5.1 TASK CONSTRUCTION

Table A.10: Results on different ratio of class split for each task in CORe50.

| $(\alpha, \beta)$ | Baseline | | TopFlow (Ours) | |
|---|---|---|---|---|
| | $A_{\text{AUC}}$ | $A_{\text{Last}}$ | $A_{\text{AUC}}$ | $A_{\text{Last}}$ |
| (100, 0) | 58.30±4.48 | 52.84±1.17 | **64.51±2.50** | **66.20±4.18** |
| (0, 100) | 87.50±1.95 | 84.52±7.79 | **87.91±2.01** | **85.20±7.08** |
| (0, 50) | **87.71±0.95** | 90.84±0.21 | 87.68±0.95 | **91.17±0.14** |
| (50, 0) | 64.36±2.42 | 76.29±8.78 | **65.46±2.16** | **77.15±7.99** |
| (50, 50) | 80.88±0.68 | 78.86±10.36 | **82.08±0.24** | **80.56±9.88** |
| (50, 100) | 80.15±2.25 | 60.61±11.25 | **81.25±1.41** | **62.79±9.03** |

**Variation of ratio of classes.** We conducted experiments by varying $\alpha$ and $\beta$, which are hyperparameters for controlling the ratio of classes that are exclusively included in specific tasks, and the ratio of classes that are included without explicit task boundaries, and the result is indicated in Table A.10. While our proposed framework, TopFlow, achieved state-of-the-art on most task configurations, the task was the most challenging when $(\alpha, \beta) = (100, 0)$, which we adopted as a main experiment configuration.

**Results on random seeds.** To further validate the reliability of the results reported in the main paper, we conducted additional experiments on CORe50 across 10 random seeds. Please refer the Table A.11. The inherent randomness of task sequences and class/domain compositions leads to high variability

across seeds. To report performance fairly, we summarize results using the mean and standard deviation. Our method consistently outperformed the baseline across all seeds, demonstrating robust improvements despite the scenario's variability. These additional experiments confirm that extending the number of seeds from 3 to 10 does not substantially change the average performance, further supporting the reliability and robustness of the proposed approach.

Table A.11: Performance comparison between MVP and Ours across different seeds.

| Seeds | MVP | | Ours | |
|---|---|---|---|---|
| | $A_{\text{AUC}}$ | $A_{\text{Last}}$ | $A_{\text{AUC}}$ | $A_{\text{Last}}$ |
| 1 | 52.82 | 51.67 | 62.01 | 55.40 |
| 2 | 59.39 | 56.07 | 59.62 | 57.22 |
| 3 | 66.25 | 70.38 | 66.59 | 71.41 |
| 4 | 57.86 | 76.50 | 58.70 | 76.73 |
| 5 | 63.04 | 65.92 | 63.46 | 67.65 |
| 6 | 58.30 | 52.84 | 64.51 | 77.00 |
| 7 | 64.20 | 75.18 | 65.69 | 78.35 |
| 8 | 67.41 | 79.43 | 68.10 | 80.45 |
| 9 | 62.78 | 54.01 | 67.01 | 66.20 |
| 10 | 62.90 | 65.04 | 63.23 | 65.18 |
| Average | 61.50 | 64.70 | 63.89 | 69.56 |
| Std | 4.40 | 10.54 | 3.12 | 8.77 |

### C.5.2 HYPERPARAMETER ABLATIONS.

We provide extended experiments on the CORe50 dataset, analyzing sensitivity to the temperature parameter $\tau$ in DFM, EMA update rate $\alpha$, $\psi$ and $r$ in GTP—the core parameters you identified as requiring deeper analysis.

Table A.12: Performance with different values of $\tau$.

| $\tau$ | $A_{\text{AUC}}$ | $A_{\text{Last}}$ |
|---|---|---|
| 0.01 | 62.94 | 65.27 |
| 0.02 | 62.62 | 65.73 |
| 0.05 | 63.09 | 65.98 |
| **0.1 (Ours)** | **64.51** | **66.20** |
| 0.2 | 60.11 | 65.56 |

Table A.13: Performance with different values of $\alpha$.

| $\alpha$ | $A_{\text{AUC}}$ | $A_{\text{Last}}$ |
|---|---|---|
| 0.9 | 63.54 | 64.52 |
| 0.95 | 64.79 | 65.48 |
| **0.99 (Ours)** | **64.51** | **66.20** |
| 0.999 | 64.03 | 65.49 |
| 0.9999 | 64.05 | 65.55 |

Table A.14: Analysis on the dimension of the $\psi$ in GTP.

| Metric | | $A_{\text{AUC}}$ | $A_{\text{Last}}$ |
|---|---|---|---|
| Baseline | | 58.30 | 52.84 |
| | 16 | 63.98 | 66.48 |
| | 32 | 64.24 | 64.85 |
| Dim of $\psi$ | **64** | **64.51** | **66.20** |
| | 128 | 64.66 | 66.02 |
| | 256 | 64.84 | 66.48 |

Table A.15: Analysis on the dimension of the $r$ in GTP.

| Metric | | $A_{\text{AUC}}$ | $A_{\text{Last}}$ |
|---|---|---|---|
| Baseline | | 58.30 | 52.84 |
| | 1 | 64.28 | 64.84 |
| | 2 | 64.20 | 65.77 |
| | 5 | 64.20 | 64.87 |
| Dim of $r$ | **10** | **64.51** | **66.20** |
| | 20 | 64.66 | 65.06 |
| | 50 | 64.39 | 66.00 |
| | 100 | 63.77 | 65.06 |

As shown in Table A.12 and Table A.13, our analysis reveals that TopFlow performs robustly across various hyperparameter settings, with performance degradation occurring only at extreme values. The optimal ranges provide stable performance, demonstrating the practical reliability. Tables

A.14 and A.15 analyze dimensionality effects of prototype $\psi$ and relationship vector $r$ in GTP. Performance improves as $\psi$ increases from 16 to 64, but degrades beyond this, suggesting that moderate dimensionality optimally balances topological representation and computational efficiency. For relationship vectors, performance enhances robustly across dimensions 1 to 100, with sub-optimal results at $r = 100$ due to overfitting instantaneous prototypes to global prototypes.

## C.6 Comparison of Computational Cost and Model Size

**Computational Cost.** We further compare the computational cost of our method with recent state-of-the-art approaches in continual learning. Table A.16 reports the total number of floating point operations (GFLOPs) required by each method, which reflects the resource overhead introduced during training. As shown in table, previous methods such as SLCA, CODA-P, and PEC introduce substantial computational burden due to additional forward/backward passes or auxiliary modules. In contrast, our method achieves superior efficiency, significantly lower than all other baselines and comparable to the baseline with only a marginal increase, while maintaining strong performance. Notably, our approach requires less than 71% of the computation needed by SLCA, and nearly 46% less than PEC, demonstrating the lightweight nature of our design. This advantage makes our method highly suitable for resource-constrained continual learning settings where efficiency is critical.

Table A.16: Comparison of Computational Cost.

| Method | GFLOPs |
|---|---|
| Baseline | 47920.083 |
| SLCA | 67150.983 |
| CODA-P | 78680.380 |
| PEC | 88476.600 |
| **Ours** | **47922.365** |

**Model Size.** Our analysis in Table A.17 confirms that TopFlow has a comparable number of parameters to the baselines, with differences within a narrow range that do not significantly impact model capacity. Therefore, the d performance improvements are not attributable to increased model size, but rather to the effectiveness of the proposed methods.

Table A.17: Comparison of parameter counts and relative increase $\Delta$.

| Method | Parameter Counts | $\Delta$ (%) |
|---|---|---|
| EWC | 85M | +0.000 |
| LWF | 85M | +0.000 |
| SLCA | 85M | +0.000 |
| PEC | 96M | +0.099 |
| CODA-P | 90M | +0.048 |
| MVP | 86M | +0.006 |
| **TopFlow (Ours)** | **87M** | **+0.018** |

# D Discussions

## D.1 Limitations and Future Works

While our proposed Online VIL setting introduces a realistic and challenging paradigm for continual learning and demonstrates competitive empirical performance, several limitations should be noted. First, our experimental evaluation is conducted on three benchmark datasets (iDigits (Volpi et al., 2021), CORe50 (Lomonaco & Maltoni, 2017), and CLEAR100 (Lin et al., 2021)) that can be reasonably adapted to the Online VIL configuration. Generalizing this framework to other widely adopted datasets presents non-trivial challenges. For example, CIFAR-100 Krizhevsky et al. (2009) lacks clearly defined domain shifts and is restricted to a single domain of natural images, which limits its applicability in multi-domain incremental learning. Datasets such as ImageNet-R Hendrycks et al. (2021) and VLCS Torralba & Efros (2011) suffer from severe class imbalance, which can lead to degenerate few-shot scenarios (1–5 samples per class) under the Online VIL setting. Similarly,

PACS Li et al. (2017) and OfficeHome Venkateswara et al. (2017) contain too few classes (7 each) to support fine-grained incremental evaluation and exhibit imbalance issues that impair consistency across tasks. As such, we highlight the pressing need to develop larger, more balanced, and richly annotated benchmarks tailored for the Online VIL scenario. Such benchmarks would enable more comprehensive evaluation and foster the advancement of continual learning methods in realistic, dynamically evolving environments.

Second, according to Algorithm A.1, Online VIL involves stochastic task construction, where the choice of random seed influences the composition and ordering of data in each task. While this design reflects the uncertainty and variability of real-world data streams, it introduces variance in experimental outcomes and poses challenges for reproducibility; for this reason, we reported the mean and standard deviation. Future work should explore robust evaluation protocols and standardized benchmark splits to mitigate these effects.

Furthermore, changes in classes, domains, and conditions can be further maximized through various modalities such as language and audio, which is a direction that needs to be studied to become more difficult and closer to the real world. In real-world deployments, learning agents are often exposed to open-world conditions, where previously unseen classes or domains may emerge without warning. The system must respond appropriately without Oracle supervision. While TopFlow demonstrates robustness in Online VIL settings with the continual evolution of known categories and domains, its performance under novel or out-of-distribution (OOD) conditions remains underexplored.

### D.2 Broader Impacts

This work tackles the Online VIL (Online Versatile Incremental Learning) scenario, where class semantics and visual domains evolve simultaneously without explicit task boundaries that reflect real-world, interactive environments more closely than traditional continual learning setups. Such dynamic learning settings are highly relevant to fields like robotics, augmented reality, and adaptive AI, where continuous real-time learning and adaptation are essential. We hope Online VIL will serve as a new benchmark for real-world incremental learning, offering a more realistic and challenging testbed for future research.

Our proposed method, TopFlow, introduces two key components:

- DFM (Domain-agnostic Flow Matching) leverages a geodesic flow-based alignment mechanism to promote domain-invariant representations, enabling models to generalize across changing visual domains. This is especially beneficial for real-world applications such as mobile robotics, autonomous driving, and field systems, where perceptual shifts (e.g., lighting, weather, terrain) are standard.

- GTP (Global Topology Preservation) maintains the global structure of the feature space over time without replaying past data, supporting lifelong learning in memory- or privacy-constrained environments. Preserving topological consistency also mitigates catastrophic forgetting and promotes safer, more stable adaptation in long-term deployment.

Despite these strengths, TopFlow's ability to learn domain-invariant features may inadvertently reduce the effectiveness of existing privacy-preserving techniques. These techniques often rely on altering visual domains (e.g., style transfer, blurring) to obscure identity. As TopFlow neutralizes domain shifts by design, it may bypass such obfuscation strategies, posing potential limitations in privacy-sensitive contexts.

### Ethics Statement

This research strictly adheres to the ICLR Code of Ethics and poses no ethical risks. We used publicly available datasets (CORe50, iDigits, CLEAR100) and model weights (pre-trained ViT), and do not include scenarios that threaten public safety, violate privacy, or cause discrimination. We only discuss real-world Online VIL scenarios and their corresponding algorithms, which are essential for AI system innovation.

### Reproduction

We have provided details on ensuring reproducibility in appropriate sections, such as 4.1 Experimental Setup in the Main Paper, A. Algorithms and Details for Methods, and B. Experimental Details in the Appendix. We also include results across multiple random seeds to demonstrate stability, and

ablation studies to clarify the contribution of individual components. The actual implementation code will be made public once the paper has been evaluated.

## LLM USAGE

We were not used to LLMs for the core methodology, ideation, scientific rigor, or originality of the research. Additionally, no LLMs were utilized in the experimental design or analysis, and all work was conducted entirely by the authors. We used LLMs only for document-level grammar checking and readability improvement.

