# OpenReview forum: "Online Versatile Incremental Learning: Towards Class and Domain-Agnostic Adaptation at Any Time"
_ICLR.cc/2026/Conference — Submitted to ICLR 2026_

### Official Review · Reviewer_bLsE · 2025-10-28

**Soundness:** 2
**Presentation:** 2
**Contribution:** 2
**Rating:** 4
**Confidence:** 5

**Summary:**

The paper proposes a new continual learning scenario called Online Versatile Incremental Learning (Online VIL), designed to better reflect real-world settings where both semantic classes and visual domains shift unpredictably and without clear task boundaries. The authors introduce TopFlow, an approach combining Domain-agnostic Flow Matching and Global Topology Preservation. Extensive experiments on several benchmarks show TopFlow achieving state-of-the-art performance.

**Strengths:**

1. The paper tackles a widely acknowledged limitation in existing continual learning setups, that the unrealistic isolation of either class increments or domain increments. This motivates the definition of the Online VIL scenario.
2. The mathematical formulations of DFM and GTP are well provided, with notation, equations, and pseudo code.
3. The proposed method achieves SOTA performance against different online continual learning and domain incremental learning methods.

**Weaknesses:**

1. The review of online CL is broad, but the paper omits explicit discussion, comparison, or empirical benchmarking against Versatile Incremental Learning approaches.
2. The experimental setup appears to be insufficiently rigorous. Specifically: The results for iDigits with varying buffer sizes are not reported. Some commonly used datasets, such as DomainNet, are not included in the experiments, limiting the comprehensiveness of the evaluation. The comparison methods differ between Table 1 and Table 2, making the evaluation inconsistent.
3. TopFlow is implemented on ViT, while several comparison methods (e.g., CBA and OnPro) are based on ResNet. Since ViT architectures generally outperform ResNet under similar conditions, the comparisons in Table 1 and Table 2 may be biased or unfair. A fairer comparison would require either architecture alignment or additional analysis to account for architectural differences.
4. This work[1] proposes methods specifically for online DIL, very closely related to the core challenge addressed by TopFlow. It should be cited and discussed in Related Work.
[1] Gunasekara N, Gomes H, Bifet A, et al. Adaptive online domain incremental continual learning

**Questions:**

Please refer to Weaknesses

---

> ### Author Response · Authors · 2025-12-04
> **Rebuttal by Authors**
>
> We sincerely thank the reviewer for their careful reading and constructive feedback.
> In particular, we appreciate the recognition that the paper addresses a significant limitation of continual learning setups and the clarity of the mathematical formulation of DFM and GTP.
> The proposed method achieves state-of-the-art performance compared to both online continual learning and domain-incremental learning baselines. Below, we address your specific concerns.
>
> ---
>
> ## **[W1] Comparison with VIL Methods**
>
> We agree that linking Online VIL with the existing VIL literature is important.
> To achieve this, we include a comparison on CLEAR100 dataset with ICON on VIL:
>
> | Method | Accuracy | Forgetting |
> |---|---|---|
> |ICON| 79.06±3.43 | 6.64±0.55 |
> | **TopFlow (Ours)** | **81.20 ± 2.75** | **6.06 ± 0.61** |
>
> and OnlineViL scenario:
>
> | Method | $A_\text{Last}$ |
> |---|---|
> | ICON | 45.15 ± 2.94 |
> | **TopFlow (Ours)** | **66.20 ± 4.18** |
>
> In Online VIL, the “VIL aspect” is limited to evolving domains and classes; under a fully streaming single-pass constraint.
> As highlighted in Section 3.1, Online VIL involves continuous, smooth distribution shifts with limited visibility, and task identifiers and multi-epoch refinements are not available.
> These features closely resemble real-world data evolution but differ from several assumptions in Standard VIL.
> For example, ICON assumes complete task visibility and explicit task transitions, which are not provided in Online VIL.
> Consequently, mechanism of ICON for storing and applying task-specific parameter shifts cannot be directly applied to an online, boundary-free stream.
> We will clarify this relationship more explicitly in the revised version and ensure that the connections between Standard VIL, Online VIL, and ICON are clearly explained.
>
> ---
>
> ## **[W2] Experimental Rigorousness (Buffer Size and Datasets)**
>
> We appreciate the suggestion to analyze the effects of buffer size and datasets.
> TopFlow is designed to be memory-free; that is, it does not depend on explicit memory management to operate.
> Our experiments show that adding a replay buffer provides only slight improvements, even with simple reservoir sampling, which supports our claim that the main benefits come from the proposed DFM and GTP mechanisms rather than from replay.
> Additionally, the scalability of Transformer architectures in terms of parameter count and data size is well understood.
> Our experiments, including Supplementary experiments with different model capacities and buffer sizes, aim to demonstrate that TopFlow remains effective across these variations rather than to optimize memory settings exhaustively.

---

> ### Author Response · Authors · 2025-12-04
> **Rebuttal by Authors**
>
> ## **[W3] Comparison with CNN-based Methods**
>
> As stated in Section B.2 of the Supplementary Material, all methods use the same ViT-base/16 backbone and the same unified training settings for consistency and fairness.
> This demonstrates that performance gains are not due to a specific architecture choice but persist even when all baselines share the same backbone and do not depend on pretraining advantages.
> This highlights the methodological contribution of DFM and GTP rather than architectural differences. We clarify this point in the main text to prevent confusion.
>
> ---
>
> ## **[W4] Relation to Gunasekara et al.**
>
> We thank the reviewer for pointing out Gunasekara et al., which we now discuss in the revised Related Work section.
> Online VIL is motivated by realistic properties of data evolution: locality and continuity.
> It simultaneously handles class expansion and domain evolution, and the smooth, local nature of these changes makes them more challenging to detect and adapt to than coarse-grained domain shifts.
> Our DFM and GTP components explicitly address these challenges by focusing on the geometry of the feature space.
> Instead of relying on explicit task signals or discrete domain boundaries, DFM and GTP align representations across layers and domains in a domain-agnostic way, which is essential for Online VIL.
>
> ---
>
> We thank the reviewer again for the insightful comments and for recognizing the strengths of our formulation and experiments.
> The added comparisons, clarifications of assumptions, and expanded discussions in the revision address your concerns and strengthen the connection between Online VIL, existing VIL methods, and broader continual learning research.

---

### Official Review · Reviewer_K7Au · 2025-10-30

**Soundness:** 3
**Presentation:** 3
**Contribution:** 4
**Rating:** 8
**Confidence:** 3

**Summary:**

This paper proposes Online Versatile Incremental Learning (Online VIL), a practical setting where both domains and classes change in a chaotic manner. Under this challenging scenario, the authors introduce two approaches to achieve efficient continual learning: (1) Domain-Agnostic Flow Matching (DFM), which extends contrastive learning to incorporate not only last-layer but also intermediate-layer features, and (2) Global Topology Preservation (GTP), which maintains the feature space through prototypes updated via online clustering. The authors evaluate their method under the designed setup on iDigits, CORe50, and CLEAR100, showing that their approach outperforms existing baselines.

**Strengths:**

1. The proposed Online VIL setting is convincing. It assumes continuous enviroment changs, unpredictable domain shifts, and the emergence of new classes, which are conditions that closely reflect real-world scenarios.
2. The proposed methodology is novel and well motivated. In particular, the contrastive operation between $h_{n,(i)}^{*}$ and $h_{n,(i)}$ in DFM is very interesting.
3. The authors provide extensive experimental evidence supporting the effectiveness of their approach.

**Weaknesses:**

1. The connection between the proposed method and the emphasized setup should be strengthened.
   1. The proposed approach does not appear to specifically target the Online VIL setting; TopFlow could also be effective under standard VIL conditions.
   2. Therefore, the authors should demonstrate that TopFlow remains effective and superior to prior methods when evaluated under the same VIL settings as previous works.
2. While the authors provide sufficient explanation that Online VIL is closer to real-world conditions, the motivation for why such a setting is necessary within this research field requires further clarification.
   1. Although the proposed setup is realistic, it would be valuable to show how and why existing methods struggle under this scenario. In the current version, it is not evident that previous methods suffer significantly in Online VIL.
   2. Without such clarification, Online VIL may appear to be an excessively challenging task at the research level. From a reviewer’s perspective, standard VIL already presents substantial difficulty, and VIL works that performs well in that setting could likely extend to real-world applications with additional engineering.
3. Including a discussion on test-time adaptation (TTA) studies would further improve the paper. The reviewer suggests that the following works be included in the 'Related Work' section.
   1. Continual learning and test-time adaptation differ primarily in whether label information is available.
   2. In the TTA field, several studies have already explored settings and methodologies similar to those addressed in this paper, including:
      1. Prototype-based adaptation [1,3]
      2. Continually changing environments [2]
      3. Sudden domain shifts and online clustering [3]
      4. Contrastive learning approaches [4,5]
   3. Additionally, the insight in Figure 2 reflects observations that have been recognized in prior literature [6,7,8]. However, the reviewer acknowledges that these findings are revisited to motivate the proposed method. Citing the following works would be beneficial.



[1] Test-Time Classifier Adjustment Module for Model-Agnostic Domain Generalization NIPS 2021

[2] NOTE: Robust Continual Test-time Adaptation Against Temporal Correlation NIPS 2022

[3] Test-time Adaptation in the Dynamic World with Compound Domain Knowledge Management RA-L 2023

[4]  Contrastive Test-Time Adaptation CVPR 2022

[5] Robust Mean Teacher for Continual and Gradual Test-Time Adaptation CVPR 2023

[6] Two at Once: Enhancing Learning and Generalization Capacities via IBN-Net ECCV 2018

[7] Revisiting Batch Normalization For Practical Domain Adaptation arXiv 2016

[8] Learning Transferable Features with Deep Adaptation Networks ICML 2015

**Questions:**

1. In Equation (6), $h$ in $h(x)$ appears to refer to the intermediate feature described in Section 3.2. Could the authors clarify this point?
2. Can the authors confirm that the features from the selected layers in Table 5 are used as $H_{n}$? Adding an explanation around Line 457 would be beneficial.
3. Including $h_{l,(j)}$ in $H_{(i)}^{-}$  is intuitively justified. However, the contribution of $h_{n,(i)}$ is less straightforward. Could the authors provide an additional ablation study?

---

> ### Author Response · Authors · 2025-12-04
> **Rebuttal by Authors**
>
> We are grateful for your thorough review and encouraging assessment of our work.
> We particularly appreciate your evaluation of the Online VIL setting as convincing, recognizing the novelty and motivation of our methodology, especially the contrastive operation in DFM, and the extensive experimental evidence supporting effectiveness.
>
> Below, we respond to your specific questions.
>
> ## **[W1] TopFlow on Standard VIL**
>
> Thank you for emphasizing the importance of evaluating TopFlow under the Standard VIL protocol.
> The Online VIL setting imposes strict streaming constraints: single-pass learning, no multi-epoch refinement, and no explicit task boundaries.
> In contrast, Standard VIL typically assumes multiple epochs per task, clear task boundaries, and the ability to store and reuse task-specific parameter updates. ICON, a method explicitly designed for Standard VIL [1], relies on multi-epoch optimization and stores task-specific parameter shifts at task boundaries.
>
> In contrast, TopFlow is designed to operate without storing task-level updates. It uses a frozen backbone and performs Domain-Agnostic Flow Matching in a way that naturally fits streaming data. To address your request, we also evaluated TopFlow under the Standard VIL protocol with the same assumptions as previous work.
>
> | Method | Accuracy | Forgetting |
> |---|---|---|
> |ICON| 79.06±3.43 | 6.64±0.55 |
> | **TopFlow (Ours)** | **81.20 ± 2.75** | **6.06 ± 0.61** |
>
> These results demonstrate that TopFlow performs at least as well as, and in some cases better than, under Standard VIL, while maintaining its robustness in Online VIL. This supports the argument that TopFlow is compatible with both settings, whereas Standard VIL methods are not readily adaptable to the online scenario.
>
> [1] Park, Min-Yeong, Jae-Ho Lee, and Gyeong-Moon Park. "Versatile Incremental Learning: Towards Class and Domain-Agnostic Incremental Learning." ECCV, 2024.
>
> ## **[W2] Necessity and detailed explanation of Online VIL**
>
> We appreciate the opportunity to clarify why Online VIL is necessary. In realistic continual learning scenarios, data arrive as a continuous stream, are often accessed only once due to storage and privacy constraints, and rarely include explicit task segmentation, such as in autonomous driving or online sensor systems. In these cases, a single-pass, boundary-free learning paradigm is needed. In Online VIL, the VIL aspect is limited to joint domain and class variation under local visibility. As discussed in Section 3.1, this leads to continuous and smooth distribution shifts that are more difficult to detect and adapt to than discrete task changes. Methods relying on clear task transitions or refinement, like ICON, are especially affected by this mismatch. To illustrate this behavior, we evaluate ICON [1] under the Online VIL protocol.
>
> | Method | $A_\text{Last}$ |
> |---|---|
> | ICON | 45.15 ± 2.94 |
> | **TopFlow (Ours)** | **66.20 ± 4.18** |
>
> These results show that ICON’s strong performance in Standard VIL relies on assumptions that no longer apply in the online scenario. In contrast, TopFlow maintains robust performance without needing task-level information or multiple passes.
> Tables A.8 and A.9 further demonstrate how changing the degree and composition of distribution shifts impacts parameter-efficient fine-tuning methods, using both $A_\text{AUC}$ (training persistency) and $A_\text{Last}$ (final performance).
> We revised the text around these tables to make the argument clearer.
>
> Additionally, we included t-SNE visualizations of the learned representations to demonstrate how feature geometry changes during Online VIL.
> Although lower-dimensional projections cannot fully capture the high-level topology, these visualizations still preserve meaningful structure and connections, underscoring the importance of Online VIL and the relevance of our approach.
>
> [1] Park, Min-Yeong, Jae-Ho Lee, and Gyeong-Moon Park. "Versatile Incremental Learning: Towards Class and Domain-Agnostic Incremental Learning." ECCV, 2024.

---

> ### Author Response · Authors · 2025-12-04
> **Rebuttal by Authors**
>
> ## **[W3] Discussion on Test-Time Adaptation (TTA)**
> Thank you for pointing out the connection to Test-Time Adaptation (TTA). We have added a broader discussion in the Related Work section to clarify the difference.
> Existing TTA methods mainly focus on adapting to domain shifts while assuming a fixed set of classes.
> Prototype-based adaptation [1] updates decision boundaries through prototypes; techniques like NOTE [2] reduce temporal correlations; and dynamic-shift approaches [3] use online clustering to monitor domain changes. Contrastive approaches [4, 5] further improve robustness, and recent continual TTA works such as BECoTTA [6] and DPCore [7] manage non-stationary domain streams, but still within a fixed, known label space.
> In contrast, Online VIL deals with both evolving domains and classes, arbitrary order of arrival, and mixed-domain batches without task boundaries, all in a fully online, label-free stream.
> The key difference is not only whether labels are available during adaptation (both Online VIL and TTA usually operate without them), but also the joint, unconstrained evolution of data across both domain and label spaces.
> We have revised the main text to clearly explain this difference and position Online VIL as a broader and more realistic setting that expands beyond the assumptions of current TTA and CTTA frameworks.
>
> Additionally, according to W3.3, we have revised the content to clarify the existing research regarding the insights in Figure 2 as follows:
>
> > While earlier studies have noted that early-layer feature statistics and normalization parameters can be sensitive to domain variation [8,9,10], these works focused on convolutional networks and conventional domain adaptation settings. Our analysis in Figure 2 revisits this phenomenon in the context of pre-trained ViTs under Online VIL, revealing a much sharper and systematic layer-wise separation: intermediate layers encode strong domain-specific structure, whereas the final layer aligns predominantly with class semantics. This layer-wise structural gap, validated by both t-SNE behavior and linear probing, highlights a representation inconsistency that becomes particularly problematic in Online VIL, where both classes and domains continually evolve. This motivates our Domain-Agnostic Flow Matching (DFM), which explicitly bridges this geometry-induced discrepancy across layers.
>
> [6] Lee, Daeun, et al. "Becotta: Input-dependent online blending of experts for continual test-time adaptation." ICML, 2024.
>
> [7] Zhang, Yunbei, et al. "Dpcore: Dynamic prompt coreset for continual test-time adaptation." ICML, 2024.
>
> [8] Pan, Xingang, et al. "Two at once: Enhancing learning and generalization capacities via ibn-net." ECCV, 2018.
>
> [9] Li, Yanghao, et al. "Revisiting batch normalization for practical domain adaptation." arXiv preprint, 2016.
>
> [10] Long, Mingsheng, et al. "Learning transferable features with deep adaptation networks." ICML, 2015.
>
> \*The reference number of the actual paper is different.
>
> ---
>
> ## **[Q1] $h$ in Equation 6.**
> To maintain generality, we chose to express it simply as $h$ in Equation 6 without specifying layer indices or using $f$.
> However, for clarity of implementation, we revised the corresponding paragraph to state this explicitly.
> In our experiments, we specifically use the final-layer features for GTP.
>
> ---
>
> ## **[Q2] Layer Selection in Table 5.**
> We appreciate the reviewer for pointing out that layer selection may appear implicit.
> To improve traceability, we revised the text around Line 457 to indicate clearly:
> The selected layers listed in Table 5 form a $(n, l)$ selection.
> This clarification will be included in the updated manuscript.
>
> ---
>
> ## **[Q3] Set of the Negative Samples.**
> Thank you for the suggestion; regretfully, it is an irremovable component of DFM.
> The contrastive objective encourages shallow features to approximate the representation manifold of deeper layers, reducing domain dependency while preserving class-relevant structure.
> This ties back to the motivation discussed in Section 3.2, where geometric flow bridges the semantic gap between early and late feature spaces.
>
> ---
>
> We thank the reviewer again for the valuable comments and for highlighting the novelty and importance of Online VIL and DFM. The additional experiments under Standard VIL, the clarified motivation for Online VIL, and the expanded discussion on TTA directly address your concerns and strengthen the overall contribution of the paper.

---

### Official Review · Reviewer_14rc · 2025-10-31

**Soundness:** 2
**Presentation:** 2
**Contribution:** 2
**Rating:** 4
**Confidence:** 3

**Summary:**

This paper introduces Online Versatile Incremental Learning (Online VIL), which simulates real-world online adaptation where both class and domain distributions shift unpredictably. The paper also introduces TopFlow, a framework that incorporates two principal mechanisms: Domain-agnostic Flow Matching (DFM), designed to induce domain-invariant representations through geometry-guided contrastive loss, and Global Topology Preservation (GTP), which preserves the global structure of the demand space online without requiring explicit data replay. Experiments demonstrate that the proposed method outperforms previous ones.

**Strengths:**

1. The introduction of Online VIL represents a step towards more realistic continual learning, simulating previously neglected, unpredictable class and domain shifts.
2. The experimental results demonstrate clear and consistent improvements across the iDigits, CORe50, and CLEAR100 datasets, with additional experiments on the challenging real-world datasets TinyImageNet and UCF-101.

**Weaknesses:**

1. The paper lacks discussion and comparison with the most recent relevant work, and contains scarcely any references from 2025/2024. The SOTA in online incremental learning algorithms should be compared to highlight the advancement of the proposed TopFlow.
2. Figure 2 is core to illustrating the justification for layering (domain vs. class focus), but the effect of GTP on feature topology is described in text only. Explicit t-SNE or clustering visualizations showing the before-and-after topology with and without GTP would better clarify this critical component and its actual effect on catastrophic forgetting and representational drift.
3. While the mathematical derivation of the DFM loss is grounded in geodesic flow kernel theory, key aspects remain underspecified for reproducibility and theoretical clarity. The normalization and batchwise projection steps in DFM are described at a high level, but implementation subtleties are omitted. For instance, how exactly the projection onto the common singular space U is computed from intermediary/final layer features and maintained for perturbed samples (particularly with only local observations) is not concretely specified.

**Questions:**

1. The paper seems to be an extension of VIL [1], expanding upon VLL to form an online VIL. What technical challenges does this extension to online learning present? How does it differ from other online learning methods? How does the paper demonstrate that its contribution is more than merely incremental?
[1] Versatile Incremental Learning: Towards Class and Domain-Agnostic Incremental Learning. ECCV 24.
2. How would TopFlow adapt to settings where domain or class axes are not explicitly separable, e.g., datasets with only a single domain (CIFAR-100) or severe class imbalance (ImageNet-R)? Is any protocol or modification suggested, or is this scenario intentionally outside scope?

---

> ### Author Response · Authors · 2025-12-04
> **Rebuttal by Authors**
>
> We thank the reviewer for the constructive feedback.
> We are especially grateful that emphasizing Online VIL as a step toward more realistic continual learning by modeling previously overlooked unpredictable class and domain shifts, and experimental results clearly and consistently show improvements across various datasets.
> We address your specific comments below.
>
> ---
>
> ## **[W1] Recent Comparisons**
>
> | Method | $A_\text{Last}$ |
> |--|--|
> | DUCT $_{\text{Mem=0}}$ [1]| 65.46±3.40 |
> | **TopFlow (Ours)** $_{\text{Mem=0}}$ | **80.64 ± 2.67** |
> | S6MOD  $_{\text{Mem=500}}$ [2]| 72.52±3.70 |
> | **TopFlow (Ours)** $_{\text{Mem=500}}$ | **90.12±1.00** |
>
> Here, “Mem” indicates the replay buffer size for CLEAR100. We have added these results to Tables 1 and 2 in the revised paper.
> These results suggest that recent state-of-the-art algorithms, while effective in their original settings, do not fully solve the challenges of Online VIL.
> DUCT [1] improves the classifier via a plug-and-play module but does not fully capture representation-level dynamics arising from continual domain and class changes.
> S6MOD [2] merges embeddings based on domain similarity, but its static merging scheme is not flexible enough for continuously changing distributions.
> We now explicitly discuss these works in the Related Work section and clarify how TopFlow addresses these limitations.
>
> [1] Liu, Sihao, et al. "Enhancing online continual learning with plug-and-play state space model and class-conditional mixture of discretization." CVPR. 2025.
>
> [2] Zhou, Da-Wei, et al. "Dual consolidation for pre-trained model-based domain-incremental learning." CVPR. 2025.
>
> ---
> ## **[W2] GTP and Visualization**
>
> Thank you for suggesting improvements to the intuitiveness of GTP
> We have added t-SNE visualizations of the feature space, providing a qualitative view of how the proposed components shape and preserve the structure of learned representations.
> Although low-dimensional embeddings inevitably lose some topological detail, the visualizations still reveal meaningful shapes and connections, aligning with our analysis of layer-wise behavior under Online VIL.
> We believe this addition makes the effects of GTP more visually accessible and complements the quantitative improvements.
>
> ## **[W3] Details for DFM**
>
> We agree that detailed algorithm descriptions are essential.
> More information about the DFM algorithm is provided in Section A.2 of the Appendix, including its formulation and implementation.
> To improve clarity, we will include explicit references in the main text whenever DFM is introduced or discussed.
> This will help readers find the full description easily without interrupting the main narrative.

---

> ### Author Response · Authors · 2025-12-04
> **Rebuttal by Authors**
>
> ## **[Q1] Distinction from Standard VIL**
>
> We appreciate the chance to clarify how Online VIL differs from Standard VIL.
> In Online VIL, the focus is on domain and class variability within an online, single-pass setting.
> As explained in Section 3.1, this setting involves continuous, smooth distributional changes with limited local visibility, which are natural in real-world scenarios but challenge many assumptions of Standard VIL.
> These assumptions do not hold in Online VIL, where tasks are not explicitly defined, each sample is seen once, and the system cannot rely on task-level storage.
> Therefore, methods primarily addressing domain shift under these assumptions often struggle in Online VIL.
> We have revised the text to emphasize these differences and to clarify why Online VIL better reflects more realistic, continual learning scenarios.
>
> ## **[Q2] Further Applicability**
>
> Thank you for the thoughtful question.
> We want to clarify that we have already analyzed situations related to class variation.
> Table 4 shows results for both the Class-Incremental Learning (CIL) and Domain-Incremental Learning (DIL), and Table A.12 provides experiments across different category distributions.
> Importantly, Online VIL inherently involves varying class exposure and imbalance, as the incoming data stream is not assumed to be class-balanced or temporally stationary.
> From this perspective, CIL/DIL and class-imbalance settings can be seen as specific cases of Online VIL.
>
> ---
> We thank the reviewer again for recognizing the real-world relevance and impact of Online VIL, and for praising our empirical results.
> The additional comparisons, visualizations, and clarifications on DFM and its differences from Standard VIL, and the discussion on label-free extensions, address your concerns and strengthen the paper.

---

### Official Review · Reviewer_dHMs · 2025-11-11

**Soundness:** 3
**Presentation:** 2
**Contribution:** 2
**Rating:** 2
**Confidence:** 3

**Summary:**

This paper addresses the challenge of continual learning under realistic, dynamic conditions where both class concepts and visual domains evolve concurrently. The authors introduce the Online Versatile Incremental Learning (Online VIL) setting to capture this scenario and propose TopFlow, a framework designed to handle such seamless transitions. TopFlow integrates two components: Domain-agnostic Flow Matching (DFM) — promotes domain-invariant representations using a geodesic flow kernel within contrastive learning. Global Topology Preservation (GTP) — maintains the global structure of the feature space without storing past samples. Experiments show that TopFlow achieves state-of-the-art results under the Online VIL setting, suggesting its potential for real-world continual learning applications.

**Strengths:**

1. This paper introduces an important problem — Online Versatile Incremental Learning (Online VIL) — a new continual learning scenario that models unpredictable and heterogeneous shifts in an online setting.

2. Extensive experiments demonstrate that TopFlow achieves state-of-the-art performance on multiple challenging Online VIL benchmarks, outperforming existing methods.

3. The work provides meaningful insights and promising directions for designing robust continual learning systems that can adapt effectively to complex, heterogeneous, and unpredictable environment shifts.

**Weaknesses:**

1. The authors propose to address the Online VIL (Online Versatile Incremental Learning) problem, but the experiments do not demonstrate the method’s effectiveness in a truly real-world online setting. Validation on a real-world application, such as autonomous driving, would make the work more compelling and impactful.

2. Most of the baselines compared in the paper, including the latest ones, are from 2024. It is recommended that the authors include more recent 2025 baselines to provide a stronger and up-to-date comparative evaluation.

3. According to ICLR guidelines, the paper should include an Ethics Statement, a Reproducibility Statement, and a declaration regarding the Use of Large Language Models (LLMs), if applicable. Adding these sections would improve the completeness and transparency of the submission.

4. It is recommended that the authors include more visualization analyses to clearly illustrate why the proposed DFM and GTP components effectively address the two main challenges of Online Versatile Incremental Learning. Such visualizations would enhance the interpretability and intuitiveness of the method.

**Questions:**

see weaknesses

---

> ### Author Response · Authors · 2025-12-04
> **Rebuttal by Authors**
>
> We appreciate thoughtful comments and positive evaluations that highlight the importance of the Online VIL problem as a new way to model unpredictable and diverse shifts.
> The recognition with TopFlow achieves state-of-the-art results on multiple challenging Online VIL tasks, and the work offers meaningful insights and promising directions for robust continual learning in complex, heterogeneous environments.
> Below, we respond to your points in more detail.
>
> ---
> ## **[W1] Real-World Validations**
>
> We understand concerns about real-world validation and scenario diversity.
> To illustrate different types of diversity, we included experiments on TinyImageNet (200 classes) and UCF-101 (101 actions) in the Supplementary Material.
> Specifically, Table A.3 reports experiments on UCF-101, where we consider wild video data with changing lighting conditions, varied camera angles, and temporal dynamics.
> This setup presents additional challenges compared to static images, especially in the temporal aspect.
> While Online VIL emphasizes gradual, continuous changes under limited visibility, video streams tend to be densely packed in time without necessarily increasing overall diversity.
> Our main focus in this work is on the theoretical and algorithmic aspects of continual learning under such online, heterogeneous shifts.
> Considering the known scalability of deep learning with respect to both parameters and dataset sizes, we selected representative yet diverse benchmarks to demonstrate how TopFlow performs across different modalities, rather than covering every possible application.
> We clarify in the main paper and Supplementary Material that these datasets serve as complementary examples of Online VIL and explain their relevance to real-world deployment scenarios.
>
> ---
> ## **[W2] Comparison with the latest SOTA**
>
> To address your request for more recent baselines, we ran experiments with DUCT [1] and S6MOD [2] and report the following results:
>
> | Method | $A_\text{Last}$ |
> |--|--|
> | DUCT $_{\text{Mem=0}}$ [1]| 65.46±3.40 |
> | **TopFlow (Ours)** $_{\text{Mem=0}}$ | **80.64 ± 2.67** |
> | S6MOD  $_{\text{Mem=500}}$ [2]| 72.52±3.70 |
> | **TopFlow (Ours)** $_{\text{Mem=500}}$ | **90.12±1.00** |
>
> Here, “Mem” represents the size of the replay buffer in the CLEAR100 dataset. Additional results have been incorporated into Tables 1 and 2 in the updated manuscript.
>
> These experiments indicate that even recent algorithms struggle to address the challenges posed by Online VIL fully.
> DUCT [1] is a plug-and-play method that mainly modifies the classifier.
> While effective in standard online continual learning, this focus makes it less suitable for capturing the evolving representation geometry under dynamic domain and class shifts in Online VIL.
> S6MOD [2] employs embedding merging based on domain differences.
> However, its static similarity-based merging strategy does not fully adapt to the continuously changing distributions in Online VIL.
>
> We now cite and discuss these methods in greater detail in the Related Work section, explicitly explaining how TopFlow differs from and complements them 2025.
>
> [1] Liu, Sihao, et al. "Enhancing online continual learning with plug-and-play state space model and class-conditional mixture of discretization." CVPR. 2025.
>
> [2] Zhou, Da-Wei, et al. "Dual consolidation for pre-trained model-based domain-incremental learning." CVPR. 2025.

---

> ### Author Response · Authors · 2025-12-04
> **Rebuttal by Authors**
>
> ## **[W3] ICLR Guidelines**
>
> We have already agreed to the ICLR Code of Ethics and LLM Usage.
> We have also provided detailed information on the reproduction environment in the Supplementary Material, and we plan to release the reproduction code once the review process is complete.
> Nevertheless, to be clear, as in the comments, we have stated the following in the Supplementary Material:
>
> > **Ethics Statement.**
> > This research strictly adheres to the ICLR Code of Ethics and poses no ethical risks. We used publicly available datasets (CORe50, iDigits, CLEAR100) and model weights (pre-trained ViT), and do not include scenarios that threaten public safety, violate privacy, or cause discrimination. We only discuss real-world Online VIL scenarios and their corresponding algorithms, which are essential for AI system innovation.
>
> >**Reproduction.**
> > We have provided details on ensuring reproducibility in appropriate sections, such as 4.1 Experimental Setup in the Main Paper, A. Algorithms and Details for Methods, and B. Experimental Details in the Appendix. We also include results across multiple random seeds to demonstrate stability, and ablation studies to clarify the contribution of individual components. The actual implementation code will be made public once the paper has been evaluated.
>
> > **LLM Usage.**
> > We were not used to LLMs for the core methodology, ideation, scientific rigor, or originality of the research. Additionally, no LLMs were utilized in the experimental design or analysis, and all work was conducted entirely by the authors. We used LLMs only for document-level grammar checking and readability improvement.
>
> ---
>
> ## **[W4] Visualizations for GTP**
> We appreciate the suggestion to make GTP more intuitive through visualization.
> In response, we have added t-SNE visualizations of the learned features.
> While low-dimensional projections cannot fully capture the high-level structure of the feature space, the visualizations offer a useful qualitative view of how representations develop and how the proposed mechanisms maintain meaningful structure and connectivity.
> We believe these visualizations make the behavior of GTP and DFM more understandable and complement the quantitative results.
>
> ---
>
> We once again thank the reviewer for emphasizing both the importance of the problem and the potential of the proposed solution.
> We hope that the added datasets, comparisons with recent methods, and clarifications regarding ethics and reproducibility address your concerns and further demonstrate TopFlow's real-world relevance and robustness.

---

### Meta-Review · Area_Chair_SDoL · 2026-01-07

**Summary:**

In this paper, the authors propose a new setting for continual learning: Online Versatile Incremental Learning, where both domains and classes change in a chaotic manner. To address the Online Versatile Incremental Learning problem, the authors introduce two approaches to achieve efficient continual learning: (1) Domain-Agnostic Flow Matching (DFM), which extends contrastive learning to incorporate not only last-layer but also intermediate-layer features, and (2) Global Topology Preservation (GTP), which maintains the feature space through prototypes updated via online clustering. The authors evaluate their method under the designed setup on iDigits, CORe50, and CLEAR100, showing that their approach outperforms existing baselines. The paper was reviewed by four expert reviewers, and three of them gave negative initial ratings before the rebuttal stage. There were some important concerns raised by the reviewers, such as the lack of comparisons with recent methods in 2025, and the motivation for why such a setting is necessary is not well-explained. The authors' responses are relatively simple, and some outstanding issues are not fully solved. For example, the authors only provide new comparisons with S6MOD and DUCT on one dataset in one setting. It seems that a major revision is required before this paper is presented at a top-tier conference.

**Reviewer Concerns:**

The paper was reviewed by four expert reviewers, and three of them gave negative initial ratings before the rebuttal stage. There were some important concerns raised by the reviewers, such as the lack of comparisons with recent methods in 2025, and the motivation for why such a setting is necessary is not well-explained. The authors' responses are relatively simple, and some outstanding issues are not fully solved. For example, the authors only provide new comparisons with S6MOD and DUCT on one dataset in one setting. It seems that a major revision is required before this paper is presented at a top-tier conference.

**Reviewer Scores:**

The paper was reviewed by four expert reviewers, and three of them gave negative initial ratings before the rebuttal stage. The authors' responses are relatively simple, and some outstanding issues are not fully solved. Only one reviewer strongly supported this paper.

---

### Decision · Program_Chairs · 2026-01-26

Reject